# A Type II-B Cas9 nuclease with minimized off-targets and reduced chromosomal translocations in vivo

Burcu Bestas [1], Sandra Wimberger [1,2,12], Dmitrii Degtev[1,12], Alexandra Madsen [1], Antje K. Rottner [1], Fredrik Karlsson[3], Sergey Naumenko[4], Megan Callahan[5], Julia Liz Touza[6], Margherita Francescatto[3], Carl Ivar Möller[1], Lukas Badertscher[1], Songyuan Li [1], Silvia Cerboni[7], Niklas Selfjord [1], Elke Ericson [1], Euan Gordon[8], Mike Firth[3], Krzysztof Chylinski[9], Amir Taheri-Ghahfarokhi [1], Mohammad Bohlooly-Y[6], Mike Snowden[10], Menelaos Pangalos[11], Barrett Nuttall [5], Pinar Akcakaya[1], Grzegorz Sienski [1] ✉ & Marcello Maresca [1] ✉

*Streptococcus pyogenes* Cas9 (SpCas9) and derived enzymes are widely used as genome editors, but their promiscuous nuclease activity often induces undesired mutations and chromosomal rearrangements. Several strategies for mapping off-target effects have emerged, but they suffer from limited sensitivity. To increase the detection sensitivity, we develop an off-target assessment workflow that uses Duplex Sequencing. The strategy increases sensitivity by one order of magnitude, identifying previously unknown SpCas9's off-target mutations in the humanized *PCSK9* mouse model. To reduce off-target risks, we perform a bioinformatic search and identify a high-fidelity Cas9 variant of the II-B subfamily from *Parasutterella secunda* (PsCas9). PsCas9 shows improved specificity as compared to SpCas9 across multiple tested sites, both in vitro and in vivo, including the *PCSK9* site. In the future, while PsCas9 will offer an alternative to SpCas9 for research and clinical use, the Duplex Sequencing workflow will enable a more sensitive assessment of Cas9 editing outcomes.

Recent advances in genome manipulations have revolutionized biomedical research, yielding numerous functional and mechanistic insights, and forming a foundation for novel medicines[1]. At the heart of the progress lay discoveries of Clustered Regularly Interspaced Short Palindromic Repeats (CRISPR)/Cas systems as a particularly effective and versatile tool for engineering genome alterations from bacteria to man[2,3]. Intense efforts on the repurposing of CRISPR-Cas systems for genome editing modifications miniaturized the complex machinery to two components: a nuclease (e.g. Cas9) and a single guide RNA (sgRNA)[4–7]. Through base-pair complementarity, the sgRNA spacer

[1]Genome Engineering, Discovery Sciences, BioPharmaceuticals R&D Unit, AstraZeneca, Gothenburg, Sweden. [2]Department of Chemistry & Molecular Biology, University of Gothenburg, Gothenburg, Sweden. [3]Data Sciences and Quantitative Biology, Discovery Sciences, BioPharmaceuticals R&D Unit, AstraZeneca, Cambridge, UK. [4]Department of Biostatistics, Harvard Chan School of Public Health, Boston, MA, USA. [5]Translational Genomics, Translational Medicine, R&D Oncology, AstraZeneca, Waltham, MA, USA. [6]Translational Genomics, Discovery Sciences, BioPharmaceuticals R&D Unit, AstraZeneca, Gothenburg, Sweden. [7]Translational Science and Experimental Medicine, Research and Early Development, Respiratory & Immunology, BioPharmaceuticals R&D, AstraZeneca, Gothenburg, Sweden. [8]Discovery Biology, Discovery Sciences, BioPharmaceuticals R&D Unit, AstraZeneca, Gothenburg, Sweden. [9]Vienna Biocenter Core Facilities, Vienna Biocenter (VBC), Vienna, Austria. [10]Discovery Sciences, BioPharmaceuticals R&D, AstraZeneca, Cambridge, UK. [11]BioPharmaceuticals R&D, AstraZeneca, Cambridge, UK. [12]These authors contributed equally: Sandra Wimberger, Dmitrii Degtev. ✉e-mail: grzegorz.sienski@astrazeneca.com; marcello.maresca@astrazeneca.com

sequence specifies the DNA target and activates the Cas enzyme for subsequent cleavage. Additionally, effective binding of the Cas-sgRNA complex to DNA requires a protospacer adjacent motif (PAM) downstream of the targeted DNA sequence[8,9]. Type II-A Cas9 orthologue from *Streptococcus pyogenes* Cas9 (SpCas9) is the best characterized Cas9 variant[10]. Due to its performance and well-characterized nature, SpCas9 was recently advanced into the first human genome editing clinical trial to treat a hereditary disease[11], with additional trials following soon after[12–15].

Two key considerations for the use of CRISPR-Cas9 system in curative treatments are their editing efficiency and their target specificity. The target specificity, also referred to as off-target effects, has recently emerged as a major liability for this system, as editing activity on both defined DNA targets, as well as other DNA sequences sharing only some degree of similarity (so-called off-targets), has been documented[16]. Previous studies have shown that SpCas9 nuclease remains active, despite the presence of up to five mismatches between the sgRNA spacer and the bound DNA[17–20]. In addition, given the difficult-to-control mutagenic activity and possible complex genomic rearrangements, the safety profile of SpCas9 has become a subject of discussion and growing concern[21–27]. Furthermore, most methods used for the evaluation of editing outcomes, including the "Verification of in vivo off-targets" (VIVO) strategy, are limited in their sensitivity by the current detection limit of targeted deep sequencing of 0.1%[28,29]. While this threshold will reliably allow detection of frequent off-target mutations, low-frequency off-targets of potential clinical significance might be overlooked during initial quality control and might generate unwanted side effects[30].

Here, to overcome these limitations we developed an off-target assessment workflow that uses Duplex Sequencing, which enabled an order of magnitude improvement in detecting Cas9 off-target mutations. Furthermore, documenting that SpCas9 off-target effects are even greater than originally thought inspired us to search for Cas9 variants with improved intrinsic high fidelity. We identified one such variant, PsCas9 and characterized its performance, showing significantly less off-target editing and chromosomal translocations when compared to SpCas9.

## Results

### Duplex Sequencing facilitates the identification of previously undetected off-target mutations induced by SpCas9

To overcome the sensitivity limit of off-target detection with NGS, we employed the highly sensitive Duplex Sequencing (Duplex-Seq). Duplex-Seq takes advantage of molecular tagging to independently barcode each DNA strand, enabling tracing of the sequence reads to each strand to distinguish and correct sequencing errors[31] (Supplementary Fig. 1). Due to the resulting extremely low sequencing error rate, Duplex-Seq facilitates the detection of base changes at frequencies as low as $10^{-9}$, thus increasing the sensitivity of mutation detection[32]. Therefore, we designed a Duplex-Seq assay to assess off-target editing of wild-type SpCas9 in vivo at the sites that have been previously analyzed by targeted amplicon sequencing[28].

We used the humanized *PCSK9* mouse model (hPCSK9-KI) we developed previously[29], and injected heterozygous hPCSK9-KI mice with adenovirus expressing wild-type SpCas9 together with the specific *Pcsk9*-targeting gMH sgRNA or with adenovirus expressing wild-type SpCas9 and GFP as a control (Fig. 1A). The gMH sgRNA was designed to be fully complementary to the mouse *Pcsk9* gene, but also targets the human *PCSK9* gene, that is present in the knock-in mice genome with a single mismatch at position 11 just outside the seed region of the spacer, resulting in simultaneous editing of both the mouse and human gene (Fig. 1B).

Out of the previously evaluated 79 off-target sites, 75 sites met technical requirements during probe design and were subjected to Duplex-Seq analysis alongside the two target sites and 37 control

regions encompassing known germline mutations and single-nucleotide variants (SNVs) present in the chosen mouse line. The average Duplex-Seq sequencing depth of 11,000× per site achieved in our experiments resulted in a sensitivity of 0.01% for the detection of mutations. We found that Duplex-Seq results of on-target gene editing on the mouse and human *PCSK9* genes analyzed with CRISPResso2[33] were very similar to the ones identified with targeted amplicon sequencing (Fig. 1C and Supplementary Fig. 2). However, in stark contrast to our previous amplicon sequencing results, the Duplex-Seq method detected events at five off-target loci with indel frequencies ranging between 0.01% and 0.04% (Fig. 1C and Supplementary Fig. 3).

Because the gMH sgRNA targets both the mouse *Pcsk9* gene and the human knock-in *PCSK9* (Fig. 1B), we also analyzed the DNA extracted from the liver tissue for the occurrence of translocations between the human and mouse *Pcsk9* sequences (Fig. 1D). To determine the frequency of these undesired translocation events after editing, we investigated the specific event where the 3′-end of mouse exon 3 translocated to human PCSK9 cDNA after CRISPR-Cas9 cleavage. We designed two Droplet Digital PCR (ddPCR)-assays detecting this event, one where the probe was placed in the mouse exon, and one where it was placed in the human region. We detected translocation events in all SpCas9-gMH treated mice with translocation frequencies ranging from 0.1 to 0.2% (Fig. 1E, F).

Taken together, due to the increased sensitivity of Duplex-Seq over the previously used targeted amplicon sequencing, we identified activity of wild-type SpCas9 on off-target sites in mouse liver that were previously missed. Additionally, we detected translocations between the two SpCas9 target sites in the mouse genome. These discoveries are highly relevant as CRISPR-induced off-targets and chromosomal translocations are potential safety risks for therapeutic editing strategies.

### Mining the human gut microbiome identified PsCas9, a new Type II-B Cas9 variant

CRISPR-Cas systems from Type II-B (FnCas9) and type V (Cas12a) have recently emerged as new genome editing tools with reduced translocations occurrences[34–39]. Because they suffer from variable and typically low activities, we set out to identify additional, efficient Cas9 enzymes that generate DNA overhangs upon DNA cleavage to minimize the risk of translocations. We used in silico approaches (Hidden Markov Models (HMM) from the TIGFRAM database) and mined the human gut microbiome for previously unannotated members of Cas nuclease family of proteins[40]. We identified a Cas operon that consists of a genomic architecture characteristic to Type II-B, harboring *cas4* and lacking *csn2* (Fig. 2A)[41,42]. The protein product of the *cas9* gene shares a high homology to WP_258333620.1 from *Parasutterella secunda* (99%), therefore we refer to it as PsCas9. Alignment of PsCas9 with all known Cas proteins classified this additional member into the type-II-B family (Fig. 2B). We further analyzed the PsCas9 operon by identifying the CRISPR repeats and trans-activating crispr RNA (tracrRNA) as before (Supplementary Data 1)[41,43]. Finally, we engineered a PsCas9 sgRNA by fusing the tracrRNA to a crispr RNA (crRNA) harboring a specific spacer sequence (Fig. 2C).

To test if PsCas9 shows DNA nuclease activity, we expressed recombinant PsCas9 protein in *E. coli* and assembled a ribonucleoprotein complex (RNP) with sgRNA[5]. We incubated the RNP with plasmids containing target DNA downstream of either randomized 3 or 4 base pairs as PAM sequences. This experimental setup allows targeting the RNA spacer to a complementary DNA sequence despite the unknown DNA recognition requirements. We observed an increase in linear or nicked DNA and a decrease in the supercoiled form of our library. PsCas9 cut the DNA in a dose-dependent manner to a similar extent in 3-nt and 4-nt libraries (Supplementary Fig. 4A). The crRNA:tracrRNA duplex harboring the PsCas9 scaffold worked specifically with PsCas9 only, and not with SpCas9 or FnCas9 (Supplementary

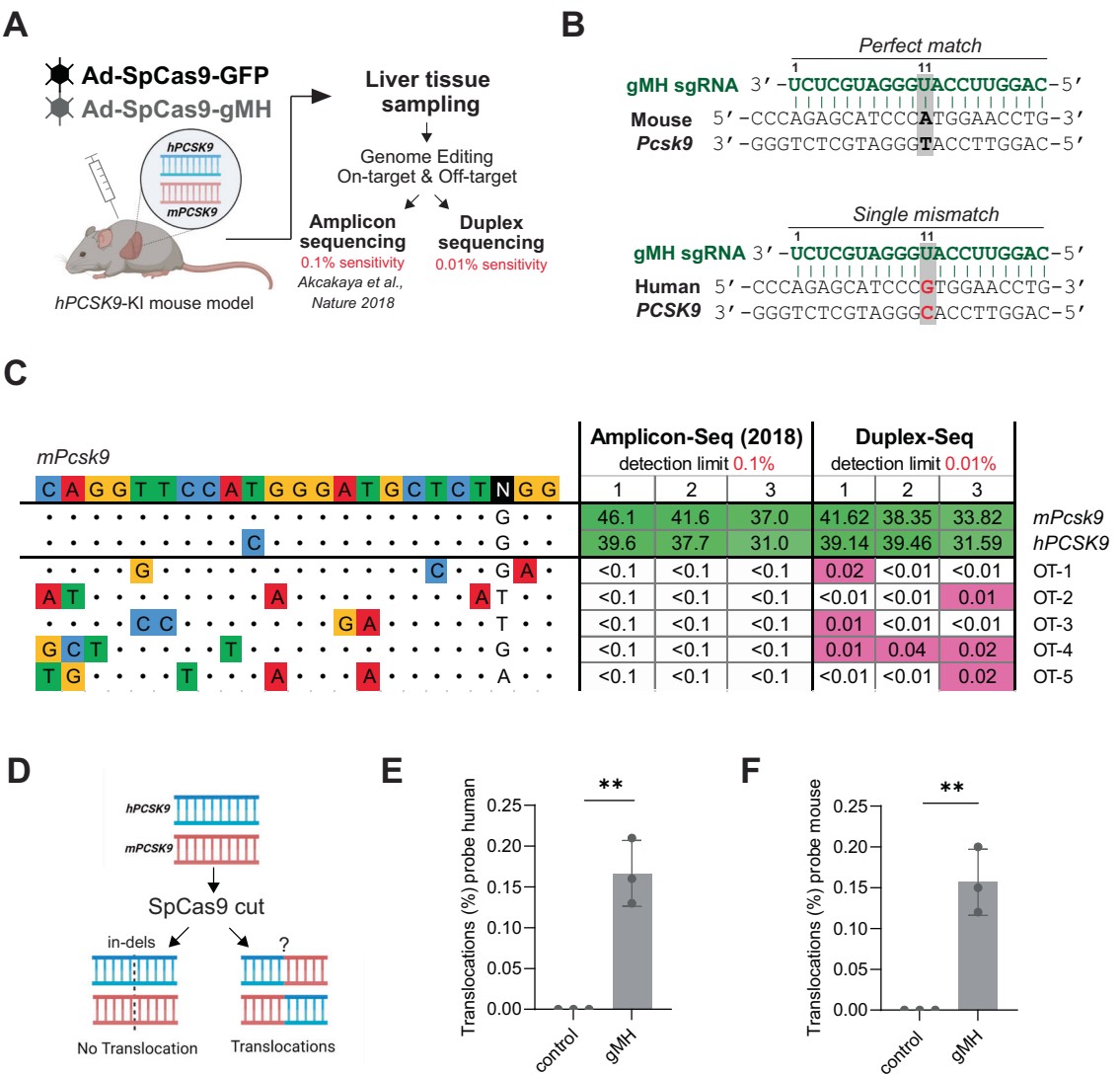

**Fig. 1 | Duplex Sequencing based analysis of SpCas9-induced off-target and translocation effects in the human *PCSK9*-KI mouse model. A** Experimental setup for SpCas9 off-target analysis in vivo using targeted amplicon sequencing and Duplex Sequencing. The humanized PCSK9-KI mouse model was injected with adenovirus expressing SpCas9 with gMH sgRNA or SpCas9 plus GFP as a control. After 1 week, the liver tissue was sampled to measure DNA editing activity. Initial off-target analysis was performed by multiplexed amplicon sequencing with a detection limit of 0.1%. Re-assessment of off-target editing at higher sensitivity (0.01% detection limit) was done by Duplex-Seq. **B** Schematic presenting the gMH sgRNA sequence targeting the mouse *Pcsk9* locus with perfect base-pair complementarity and the human *PCSK9* locus with a single mismatch at position 11 (highlighted in red). **C** Comparison of amplicon sequencing (Amplicon-Seq) vs. Duplex-Seq-based off-target analysis. Gene editing analysis at off-target sites[28] by amplicon sequencing or Duplex-Seq followed by CRISPResso2 analysis. Heatmaps present the editing efficiency for the two target sites (highlighted in green), as well as the five detected off-target sites (highlighted in magenta). Amplicon sequencing and Duplex-Seq were performed with *n* = 3 biologically independent mice (1, 2, 3) using genomic DNA isolated from the liver of mice treated with adenoviral vector expressing sgRNA gMH together with SpCas9. DNA from the same mice (numbered 1–3) were used for both sequencing approaches. Values show editing efficiency as percent indel-containing reads normalized to mice treated with adenoviral vector encoding SpCas9 without an sgRNA as a control. Extended tables showing non-normalized data of all assayed off-target sites, as well as the two target sites for both amplicon sequencing and Duplex-Seq can be found in Supplementary Figs. S2 and S3, respectively. The detection limits reached were 0.1% and 0.01% for amplicon sequencing and Duplex-Seq, respectively. **D** Schematic representation of the possible outcomes of the gene editing at the *Pcsk9* locus in humanized PCSK9-KI mouse model. **E, F** SpCas9-induced translocations between mouse and human *PCSK9* genes after in vivo editing. Frequency of the specific translocation event where the 3' end of mouse exon 3 becomes fused to the 5' end of human exon 3 assayed by ddPCR. In (**E**), the probe was placed in the human exon, whereas in (**F**) the probe was placed in the mouse exon. Graph shows individual values, as well as mean ± SD from *n* = 3 biologically independent mice per group. A two-sided Student's *t* test was performed to evaluate statistical significance (**P value = 0.002 (**E**) and 0.0026 (**F**)).

Fig. 4B). We concluded that PsCas9 acts as a DNA nuclease in an RNA-dependent manner.

To identify the preferred PAM sequence of PsCas9, we employed the 4-nt randomized PAM plasmid library and analyzed cleaved products by amplicon sequencing (Fig. 2D). The enrichment of each identified PAM variant was normalized to the frequency in the initial library and evaluated by a position frequency matrix (PFM)[44]. Our

analysis revealed that PsCas9 shows a strong preference for NGG PAM (Fig. 2D), which is similar to the well-studied SpCas9[5]. To determine the DNA cleavage pattern of the protospacer sequence, we examined the amplicon sequencing data. PsCas9 preferentially cleaved the target strand 3 nucleotides upstream of the PAM, while the non-targeted strand was cleaved 6-7 nucleotides upstream of the PAM (Supplementary Fig. 4C). This suggests that the PsCas9-mediated cleavage

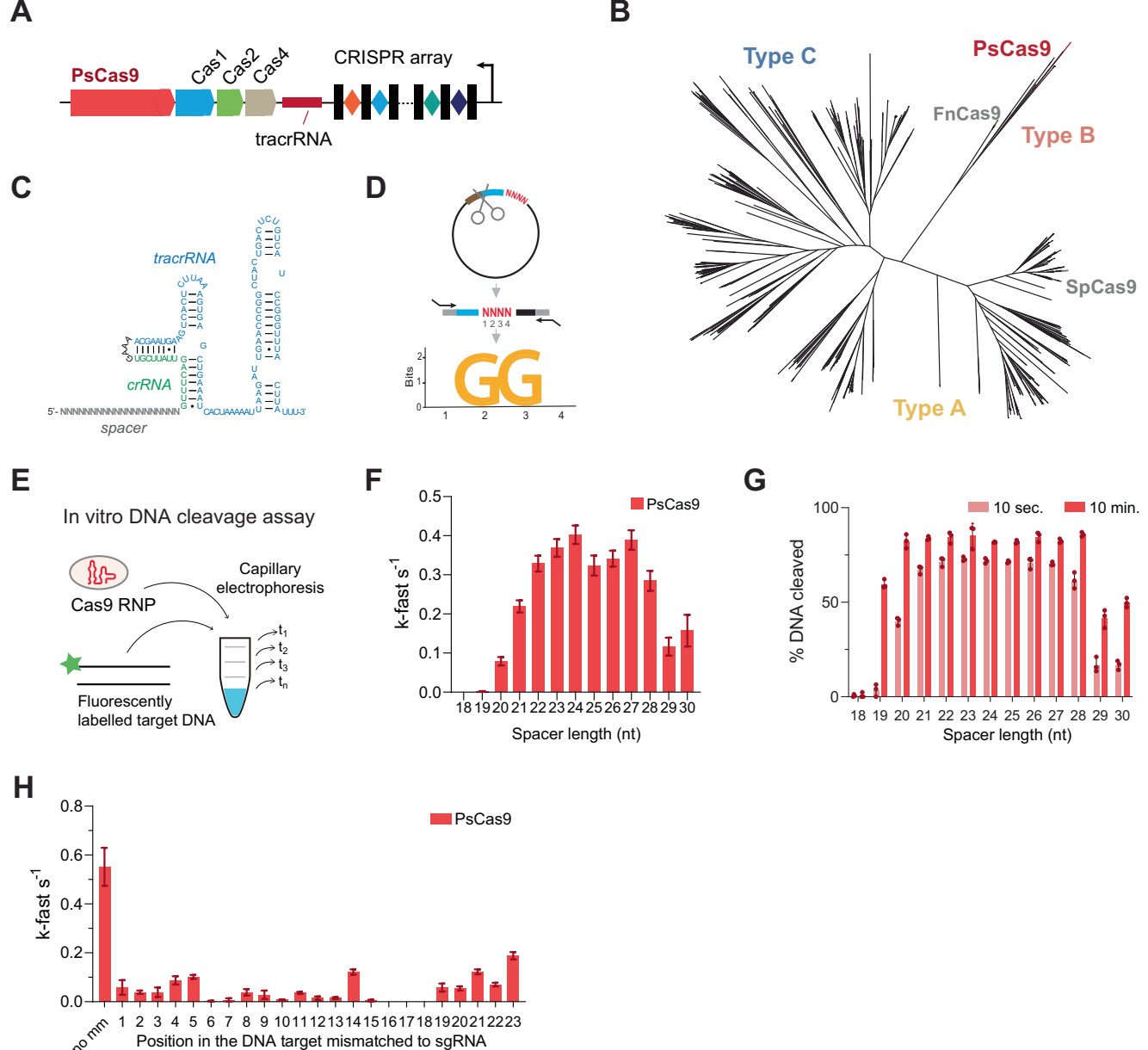

**Fig. 2 | Identification and characterization of PsCas9, an RNA-guided DNA nuclease. A** Schematic representation of PsCas9 CRISPR locus. Red line: tracrRNA; black rectangles: repeats; diamonds: spacers; black arrow: transcription direction of repeat–spacer array. Note that for simplicity repeat-spacer arrays do not represent the actual number of spacers. **B** Phylogenetic tree of Cas9 orthologs with depicted Type II-A, II-B and II-C systems (data from[53]). **C** Secondary structure of the optimized sgRNA scaffold for PsCas9 visualized by CLC Genomics Workbench. CRISPR repeat:tracrRNA modules are colored in green and blue respectively. **D** Experimental approach to identify the preferred PAM sequence of PsCas9. A plasmid library consisting of randomized four base pair PAM sequences (red) downstream of a defined target sequence (blue/brown) was used, followed by NGS analysis of the cleaved products. The preferred PAM sequence was calculated by position frequency matrix (PFM) and identified as NGG PAM. The sequence logo was visualized with R using the ggseqlogo package. **E** In vitro DNA cleavage assay performed by incubating RNP complexes with fluorescently labeled DNA substrate. **F**, **G** In vitro DNA cleavage of the *EMX1a* substrate using sgRNAs with spacer length ranging from 18 to 30 nucleotides (see Supplementary Fig. 4D). Presented are k-fast s$^{-1}$ (**F**) and % DNA target cleavage after 10 s and 10 min reaction, respectively (**G**). Data are presented as mean ± SD, $n = 3$ (all experimental data points available in Source Data). **H** In vitro DNA cleavage rates of *EMX1a* with a mismatch at every single base position along the protospacer (see Supplementary Fig. 4G). Mismatch positions are labeled in 3′ to 5′ direction, with position 1 being directly upstream of the PAM. "no MM" = no mismatch. Data are presented as mean ± SD, $n = 2$ (all experimental data points available in Source data).

generates predominantly 3-nt long 5′-overhangs in the DNA substrate, unlike SpCas9, which leaves 1-nt overhangs or a blunt-ended cut[45–50]. Overall, we identified PsCas9 as an additional member of the Type II-B Cas9 nuclease family, and showed that its nuclease activity generated 3-nt long 5′-overhangs in the DNA substrate. This also highlights the value of human microbiome mining for the discovery of Cas9 nucleases with different activity profiles that may overcome the limitations of existing Cas9s.

## PsCas9 performs best with ≥22-nt spacers and requires specific sgRNA-target pairing

Given the qualitative difference in DNA cleavage by PsCas9 and SpCas9, we next compared the kinetics of DNA cleavage catalyzed by the two Cas nucleases. To this end, we performed an in vitro time course cleavage assay by incubating Cas9 RNP complexes with fluorescently labeled DNA substrates (Fig. 2E). The reactions were quenched at specific time points over the course of 10 min and digested

DNA was analyzed by capillary electrophoresis[51,52]. The fraction cleaved substrate was calculated and fitted to a bi-exponential decay function[51]. Speed constant k_fast, a single parameter largely describing the reaction rate, was extracted and compared for each substrate–enzyme pair.

As the activity of many Cas9-enzymes depends on the length of the sgRNA spacer[53], we systematically optimized the spacer length for PsCas9 sgRNA. We measured the cleavage kinetics with a series of sgRNAs ranging from 18 to 30 nucleotide spacer length targeting fluorescently labeled synthetic *EMX1a* target (Supplementary Fig. 4D). In contrast to SpCas9[54], PsCas9 catalyzed DNA cleavage at high speed and efficiency with a spacer length of 22–28 nucleotides (Fig. 2F, G). We then re-evaluated PsCas9 cleavage kinetics with 20- and 22-nt spacers at two additional DNA substrates, AAVS1 and CD34, and compared it with SpCas9. PsCas9 cleaved DNA substrates at a similar rate and efficiency to SpCas9 when using a spacer of optimal length (Supplementary Fig. 4E, F).

Subsequently, we used the cleavage kinetics as an in vitro assay to establish the sensitivity of PsCas9 to mismatches between the sgRNA spacer and the targeted DNA. To this end, we designed a series of labeled DNA substrates where we systematically disrupted the base paring of the sgRNA to the target DNA at each position along the DNA sequence (Supplementary Fig. 4G). Strikingly, we found that mismatches in the DNA substrate largely decreased the cleavage speed of PsCas9 across the target DNA sequence, particularly at positions 6-13 and 15-18 (Fig. 2H). In contrast, SpCas9 tolerates mismatches readily across the spacer, a property observed previously and postulated to explain SpCas9 off-target activity (Supplementary Fig. 4H)[17–20,55]. Both enzymes cleave the perfect substrate (no mm) almost to completion within 10 seconds ((Supplementary Fig. 4I, J). However, PsCas9 requires up to 10 minutes to process mismatched DNA while SpCas9 is capable of significantly faster digestion for most substrates ((Supplementary Fig. 4I, J). Overall, our data indicates that for efficient and rapid cutting, PsCas9 requires a high level of complementarity to its DNA substrate and thus displays a high intrinsic specificity.

### PsCas9 programmed by an sgRNA acts on the human genome in a sequence-dependent manner

To explore whether PsCas9's ability to cut DNA in vitro would also translate to genomic DNA in mammalian cells, we codon-optimized its coding sequence and engineered nuclear localization signals. We transfected HEK293T cells with plasmids encoding FLAG-tagged PsCas9, SpCas9, or FnCas9 and confirmed comparable expression levels of all proteins by Western blot (Fig. 3A).

Next, we tested PsCas9's activity on human genomic DNA by co-expressing PsCas9 and a U6 promoter-driven sgRNA expression cassette containing its specific sgRNA scaffold in the same cell. We directed PsCas9 to target endogenous loci *EMX1a* and *CD34* with sgRNAs of variable spacer lengths in HEK293T cells and analyzed the targeted DNA sequences by NGS. In line with our in vitro results, PsCas9 cuts genomic DNA as evidenced by the accumulation of DNA insertions and deletions (indels, hereafter referred to as genome editing), at both targeted loci (Fig. 3B). Notably, PsCas9 showed the highest activity with 21-22 nt spacer length, consistent with in vitro observations.

Subsequently, we compared the activity of PsCas9 and SpCas9 at four genomic sites (*CD34*, *STAT1*, *AAVS1*, *HBEGF*). Interestingly, while the targeted DNA sites showed rather similar genome editing regardless of the nuclease (Supplementary Fig. 5A), the underlying mutation pattern of each nuclease was different. As previously observed[45–50], SpCas9-guided genome cutting elicited 1-nt insertion on most sites (Supplementary Fig. 5B). In contrast, PsCas9 predominantly facilitated the insertions of 3-nt, which is in line with our in vitro observations of 3-nt overhangs after DNA cleavage (Supplementary Fig. 5B). This suggests that PsCas9 cuts genomic DNA in a manner agreeing with in vitro

results and creates DNA overhangs which are likely substrates for the non-homologous end joining (NHEJ) DNA repair pathway[56]. Altogether, we conclude that PsCas9 can be used to engineer the human genome. While its efficiency is quantitatively comparable to SpCas9, PsCas9's editing outcome is qualitatively distinct.

### PsCas9 promotes directional DNA insertions into the human genome

We wondered if PsCas9 can be used to introduce DNA(perform knock-ins) into the human genome in a similar way as shown in vitro (Supplementary Fig. 5C). For this, we first designed blunt-ended dsDNA oligos (34-nt in size[16]) to insert into the genome after introducing a double-strand break with either PsCas9 or SpCas9. At two independent loci (*CD34* and *STAT1*), we observed similar efficiencies of knock-in ((Supplementary Fig. 5D, E). We then repeated the experiment with 3 nt-long 5'-overhangs complementary to the target site added to the dsDNA oligos. Strikingly, PsCas9 but not SpCas9 leads to increased efficiency and directionality of the knock-in with the use of the DNA oligos with 5'-overhangs(Supplementary Fig. 5D, E).

To understand the mechanism of PsCas9 dsDNA insertions, we utilized small molecules to modify the DNA repair process. As similar DNA lesions are repaired by NHEJ[16,56,57], we repeated the knock-in experiments with a small molecule inhibitor of the NHEJ-specific DNA-dependent protein kinase (DNA-PKi, M9831)[58]. The treatment with DNA-PKi abrogated DNA oligo knock-in regardless of the presence of 5'-overhangs or Cas nuclease, proving the mechanism of action to be NHEJ (Supplementary Fig. 5D, E). In summary, PsCas9-mediated cuts in the human genome likely leave 5'-overhangs that could be used for directional knock-ins mediated by NHEJ.

We next tested if PsCas9-mediated genome editing supports HDR-based genome insertions. For this, we chose to install the F508del mutation in *CFTR* (deleting CTT), R14del mutation in *PLN* (deleting AGA) and E342K mutation in *SERPINA1* (G to A, two different sgRNAs). We designed single-stranded oligonucleotide templates containing the mutated sequence of interest flanked by 50-nt homology arms to the respective locus. At the three independent loci targeted by PsCas9 or SpCas9, we observed similar or higher efficiencies of precise HDR-mediated knock-in for PsCas9 (Supplementary Fig. 5F). We conclude that the 5'-overhangs after the PsCas9 cut do not impair the genome integration via the HDR process.

### PsCas9 genome editing activity requires stringent base-pair complementarity with the targeted genomic site

Our in vitro experiments suggested that PsCas9 is sensitive to mismatches between the DNA target and the sgRNA spacer. In order to evaluate how PsCas9 tolerates imperfect base pairing of the sgRNA in cells, we targeted the *EMX1a* site with a 23-mer spacer, which previously showed efficient genome editing (Fig. 3B), and mismatched every position individually to the complementary base. Amplicon sequencing analysis revealed that the genome editing elicited by PsCas9 is drastically reduced by mismatches between positions 1-16 distal to the PAM (Fig. 3C). This suggests that PsCas9 has more conserved sequence requirements and less tolerance to mismatches near the seed region compared to SpCas9[18,20,59–61]. Collectively, our in vitro and cellular studies demonstrated that PsCas9 requires large complementarity to its target DNA, and therefore is a high-specificity DNA nuclease.

### PsCas9 discriminates on-target sites from candidate off-target sites in mammalian cells

Reasoning that PsCas9's high sensitivity to mismatches would result in target-specific cutting, thus presumably low activity on candidate off-target sites, we benchmarked PsCas9 with wild-type SpCas9 and FnCas9, a Type II-B Cas9 variant of high specificity[36]. Given that all

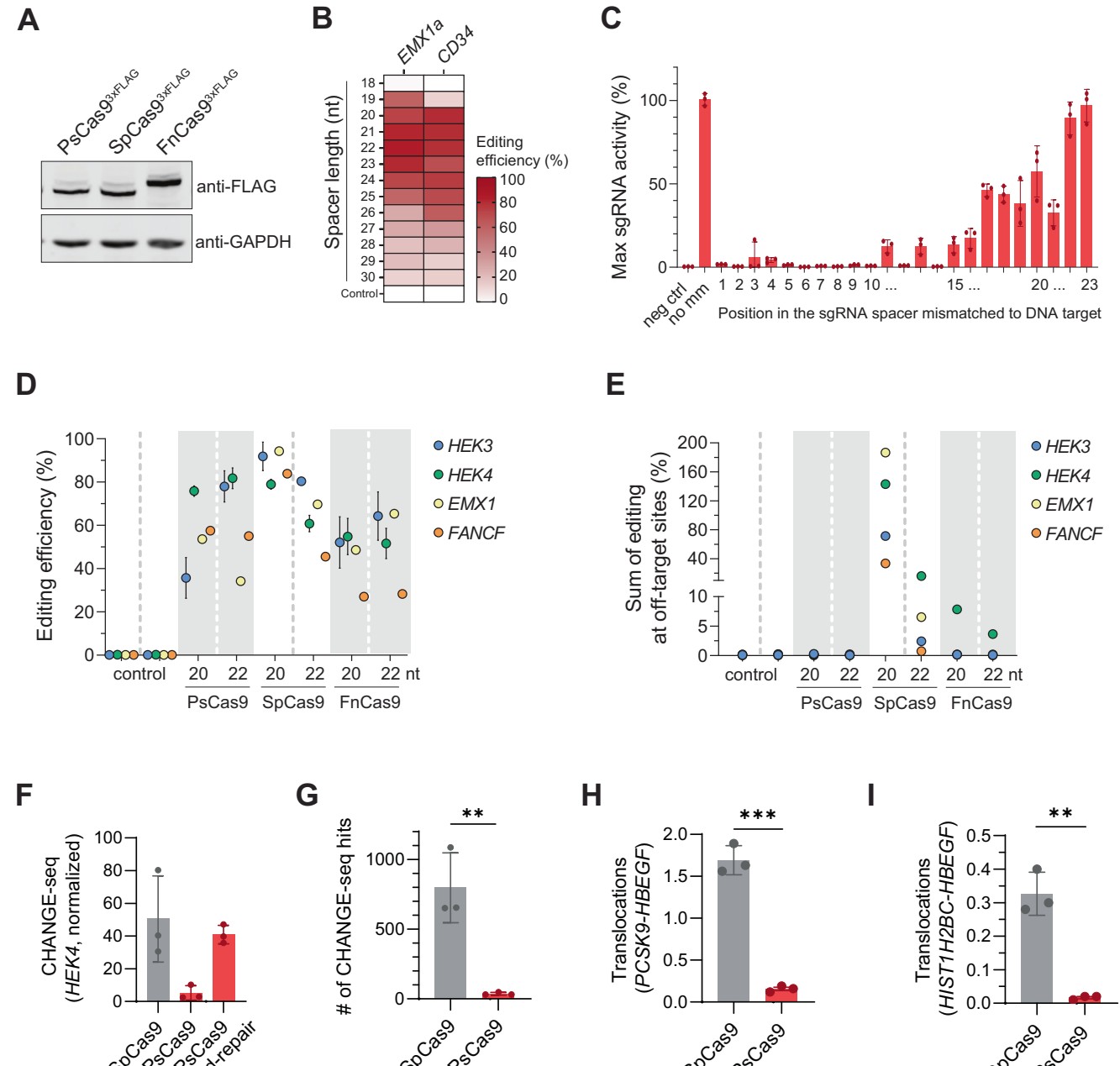

**Fig. 3 | PsCas9 edits the human genome and shows low off-target activity.**
**A** Western blot analysis using an antibody detecting the FLAG-tag of PsCas9,
SpCas9 and FnCas9 expressed in HEK293T cells. GAPDH was used as a loading
control. **B** Heatmap showing the editing efficiency in HEK293T-cells transfected
with PsCas9 and sgRNAs with various spacer length (18-30nt) targeting either
*EMX1a* or *CD34*. The editing efficiency (percent modified reads per sample) was
calculated using amplicon sequencing with CRISPResso2 analysis. Data are pre-
sented as mean, *n* = 3. **C** Maximum activity of sgRNAs (%) targeting *EMX1a* con-
taining a single mismatch at the indicated position in HEK293T cells. The data was
normalized to the sgRNA with a spacer fully complementary to the *EMX1a* target
site. The editing efficiency was calculated with CRISPResso2 analysis of amplicon
sequencing data and shown as percent modified reads in each sample. 5′ of sgRNAs
contain a G-G mismatch at position 23 that comes from U6 transcription. Data are
presented as mean ± SD, *n* = 3. **D, E** On-target editing (**D**) and off-target editing (**E**)
by PsCas9, SpCas9 and FnCas9 with 20 nt and 22 nt spacers at the depicted loci. For
the off-target analysis, the sum of editing at three to four off-target sites was used

(See Supplementary Fig. 6A). Control samples were untransfected and the editing
efficiency was determined using NGS. Data was analyzed by CRISPResso2 and
presented as percent modified reads in each sample. Data are presented as
mean ± SD, *n* ≥ 3. **F** CHANGE-seq read counts after in vitro genome cleavage with
SpCas9, PsCas9 or PsCas9 with optimized CHANGE-seq protocol (PsCas9 end-
repair) at the *HEK4* site. The signal was scaled to the library depth (shown as read
counts per million mapped reads). Data are presented as mean ± SD, *n* = 3. **G** Bar
plot showing the number hits identified with CHANGE-seq using HEK4 sgRNA for
SpCas9 and PsCas9. Displayed hits are present in at least one technical replicate
(out of three) for each nuclease. Data are presented as mean ± SD, *n* = 3 (unpaired *t*
test, two-tailed; **P* value = 0.0020). **H, I** Translocation rates for SpCas9 (gray) and
PsCas9 (red) between the depicted targeted loci in HEK293T cells. Balanced
translocations were measured by custom ddPCR assays detecting the predicted
translocation events. Data are presented as mean ± SD, *n* = 3 (unpaired *t* test, two-
tailed; **P* value = 0.0011; ****P* value = 0.0001).

enzymes share the same NGG PAM, we selected sites with previously identified SpCas9 off-targets (H*EK4, EMX1, FANCF* and *HEK3*)[20]. As we found in in vitro experiments, PsCas9 requires a 22-nt spacer for optimal activity, while SpCas9 and FnCas9 are usually used with 20-nt spacer guide RNA. Due to these distinct preferences, we compared on- and off-target editing activity of the three enzymes with both 20- and 22-nt spacers. While PsCas9 showed on-target editing across all targeted sites, albeit to various degrees, the editing at off-target sites was very low, typically at or close to background levels (Fig. 3D, E and Supplementary Fig. 6A). Wild-type SpCas9 showed high activity at on-target sites as well as at many off-targets. FnCas9 was editing the targeted sites with comparable or lower efficiency than PsCas9 across conditions, but with a higher frequency at off-target sites (Fig. 3D, E and Supplementary Fig. 6A). Compared to SpCas9, the on-target editing of PsCas9 was lower at *EMX1* and *FANCF* target sites (Supplementary Fig. 5A), suggesting differences in locus and sequence specific editing properties. Altogether, we found that the strict requirements of PsCas9 with specific and unperturbed base pairing between sgRNA and DNA target vastly reduce its unintentional activity at closely resembling genomic loci.

## PsCas9 demonstrates less genome-wide off-target cleavage activity in vitro

PsCas9 is a Type II-B family enzyme with distinct properties, thus its genome-wide off-target activity might differ from SpCas9. We therefore aimed to analyze the genome-wide off-target profile of PsCas9 with a sensitive orthogonal method, CHANGE-seq[62]. We modified CHANGE-seq to capture PsCas9 staggered ends and demonstrated the modified CHANGE-seq provides on-target cleavage detection for PsCas9 at a higher efficacy compared to original CHANGE-seq and at a similar efficacy to SpCas9 (Fig. 3F and Supplementary Data 7). To evaluate genome-wide off-targets of PsCas9 and SpCas9, we used a previously reported promiscuous SpCas9 sgRNA, *HEK4*[62] on human genomic DNA. The *HEK4* sgRNA spacer was extended to 22-nt for optimal PsCas9 activity. We identified many off-target cleavage sites for SpCas9 in vitro: 1087, 655 and 651 sites in three technical replicates respectively, with 372 shared sites across all replicates (Fig. 3G and Supplementary Data 7). For PsCas9, the number of detected off-target cleavage sites was substantially lower: 49, 31, and 31 sites in three technical replicates respectively, with six shared sites across all replicates (Fig. 3G, Supplementary Fig. 5B, and Supplementary Data 7). All six shared PsCas9 off-target sites contained NGG PAM. Four of them were identified in all SpCas9 replicates and had 2 or 3 mismatches in the spacer sequence. One site was identified in 2 SpCas9 replicates and harbored 4 mismatches, while one site was identified in one SpCas9 replicate and harbored 1 bulge and 1 mismatch. The sites identified in only one or two replicates generally had low CHANGE-seq read counts (overall mean 6.7 counts per hit) and accounted for less than 10% of the on-target read counts (Supplementary Data 7), consistent with the possibility that these sites may be detected by chance because of the assay limit of detection. Taken together, our results demonstrate that PsCas9 generates much fewer genome-wide off-targets in vitro compared to SpCas9.

## PsCas9 induces fewer translocations after gene editing in mammalian cells

Chromosomal translocations may occur after induction of multiple, simultaneous DSBs, e.g. between on- and off-target sites cleaved by promiscuous enzymes. Such genomic instability poses a safety liability for using nucleases in clinical applications. Therefore, we investigated if the 5′-DNA overhangs after cleavage by PsCas9 are as translocation-prone as the blunt ends typically generated by wild-type SpCas9. To this end, we transfected HEK293T cells with either SpCas9 or PsCas9 and a pair of sgRNAs previously shown to induce balanced translocations between the two on-target sites by SpCas9[63,64]. Using ddPCR, we found that PsCas9 induced 22-fold- and 11-fold fewer translocations than SpCas9 between *HIST1H2BC* and *HBEGF*, and between *PCSK9* and *HBEGF*, respectively (Fig. 3H, I). Our results demonstrate that PsCas9 is less prone than SpCas9 to induce translocations after introducing two independent DSBs in the human genome.

## In vivo gene editing by PsCas9 shows less off-target editing and reduced chromosomal translocations

With its favorable intrinsic specificity, PsCas9 could be an attractive candidate enzyme for safe and precise genome editing. To further assess its editing activity in vivo, we used the heterozygous hPCSK9-KI mice transduced with adenovirus containing PsCas9 together with the gMH sgRNA (Fig. 4A), and collected blood samples and liver tissue one week post-injection to assess the outcome of Cas9 nuclease gene editing activity. We compared gene editing of the two target sites in PsCas9-treated and SpCas9-treated animals (from Fig. 1A) by amplicon sequencing. Interestingly, we found comparable efficiency of PsCas9 and SpCas9 in generating indels at the perfectly complementary mouse *Pcsk9* locus (Figs. 4B and 1C). Notably, even though both nucleases induced similar levels of editing, the corresponding protein reduction in PsCas9-treated animals was lower compared to SpCas9 (Supplementary Fig. 7A). As anticipated, the nature of predominant +3 insertions induced by PsCas9 resulted in less frequent out-of-frame deletions compared to SpCas9 in the mice, presumably leading to the observed lower plasma protein reduction (Supplementary Fig. 7B). Importantly, while SpCas9 showed similar on-target efficiency at both human and mice loci, PsCas9 edited the human *PCSK9* locus at ~40% lower efficiency than the mouse *Pcsk9* locus despite the rU-dG mismatch that is often tolerated due to wobble base pairing (Fig. 4B)[65]. This improved fidelity in vivo aligns with our observations of the increased mismatch tolerance in vitro and in HEK293T cells for the *EMX1a* targeting guide RNA.

In order to further validate the higher specificity of PsCas9 compared to SpCas9, we applied our Duplex-Seq pipeline for off-target analysis to the PsCas9-treated mice. In line with our cell line data, no off-target editing could be detected in any of the three PsCas9-treated animals (Fig. 4C and Supplementary Fig. 3). Collectively, our in vivo results further confirmed the high specificity of PsCas9.

We wondered if the 5′ DNA overhangs after PsCas9 cut are as translocation-prone as the one generated by wild-type SpCas9 in vivo. By extracting reads from the Duplex-Seq data and mapping them to both target sites, we analyzed the frequency of these translocations between mouse and human *PCSK9* loci (Fig. 4D). Quantification of those reads as a fraction of total reads at the two target sites revealed the presence of the translocations in animals edited with wild-type SpCas9 (Fig. 4E, F). A similar analysis of the PsCas9-treated animals showed a significantly reduced frequency of chromosomal translocations as compared to SpCas9 (Fig. 4E, F). Of note, both enzymes showed a similar distribution of detected translocation types, with dicentric and acentric translocations being the most common (Supplementary Fig. 7C, D). Taken together, our in vivo results corroborate the in vitro data and altogether suggest the high intrinsic specificity and safety of PsCas9 for genome editing.

## Discussion

The advent of CRISPR-Cas systems has opened opportunities to manipulate and modify the human genome at will. This has led to an enormous amount of interest in developing CRISPR-Cas systems into safe, effective, and curative treatments for individuals suffering from genetic diseases. The key considerations for newly emerging genetic medicines are their highly controlled and predictable action to avoid unexpected adverse effects. For example, induction of potentially pathogenic mutations at sites that are highly similar to the edited one, i.e. off-target editing, and the induction of chromosomal alterations such as translocations are of major concern for use of gene editing in a

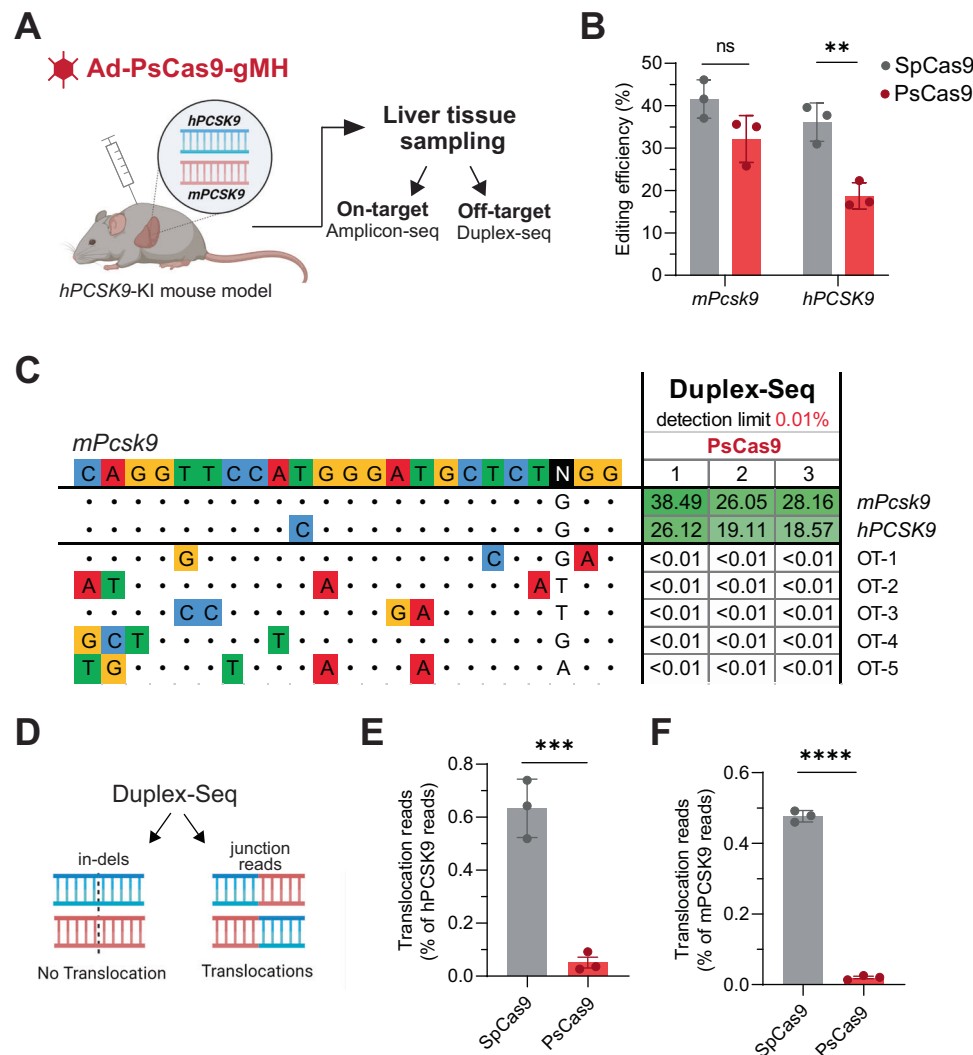

**Fig. 4 | In vivo gene editing with PsCas9 in the human *PCSK9*-KI mouse model.**
**A** Experimental setup for the in vivo workflow. Humanized PCSK9-KI mice were injected with adenovirus expressing PsCas9 and gMH sgRNA. After 1 week, liver tissue was sampled to estimate DNA editing activity. On-target editing was analyzed by targeted amplicon sequencing, off-target editing by Duplex-Seq. **B** The editing efficiency at mouse *Pcsk9* and human *PCSK9* elicited by PsCas9 and SpCas9 as determined with amplicon sequencing followed by CRISPResso2 analysis is shown as percentage of indel-containing reads in each sample. Graphs show individual values, as well as mean ± SEM from n = 3 biologically independent mice. A two-way ANOVA followed by Sidak's multiple comparison test was performed to evaluate statistical significance (**P value = 0.009; ns P value = 0.1846). **C** Gene editing at off-target sites assayed by Duplex-Seq followed by CRISPResso2 analysis. Heatmaps present editing efficiency for the two target sites (highlighted in green), as well as the five off-target sites detected in SpCas9-treated mice (Fig. 1C). Duplex-Seq was performed with n = 3 biologically independent mice (1, 2, 3) using genomic DNA isolated from the liver of mice treated with adenoviral vector expressing sgRNA

gMH together with PsCas9. Adenoviral vector encoding SpCas9 without an sgRNA was used as a control. Values show editing data as percent indel-containing reads normalized to controls. An extended table showing non-normalized data of all assayed off-target sites, as well as the two on-target sites can be found in Supplementary Fig. 3. The detection limit reached in this experimental set-up was 0.01%. All Duplex-Seq experiments were performed in parallel in the same experiment. **D** Schematic of using Duplex-Seq to assay indels and junction reads that report on translocation frequencies. **E, F** Evaluation of translocations between the two *PCSK9* on-target sites in vivo. Translocation percentage shown as the fraction of translocated sequencing reads divided by all human (**E**) or all mouse on-target reads (**F**). Graphs show individual values, as well as mean ± SD from n = 3 biologically independent mice. Unpaired, two-tailed t test was performed to evaluate statistical significance (***P value = 0.0010 (**E**); ****P value < 0.001 (**F**)). Note that all four possible translocation outcomes between the two genes are included in the analysis.

therapeutic setting. Accounting for these concerns, the FDA has recently released the draft guidance for the safe development of new genetic medicines that recommends the use of "methods with adequate sensitivity to detect low frequency events" as a necessary step in validating safety[66].

In the present study, we show that the most commonly used standard NGS-based methods for Cas9 off-target detection might underestimate the safety risks associated with therapeutic gene editing. In general, NGS-based mutation detection is typically limited to a sensitivity of 0.1% and off-target indels occurring at 0.1%, although low, could affect thousands of cells in the context of an adult human tissue.

In the context of PCSK9-lowering gene therapy to treat a large population of hypercholesterolemic patients for example, this could lead to unpredictable side effects. To address this concern, we have now applied a more sensitive Duplex-Seq workflow to re-assess Cas9 off-target effects in vivo. To the best of our knowledge, this is the first example of Duplex-Seq for Cas9 off-target assessment. Our results highlight that Duplex-Seq provided a 10-fold improved sensitivity, resulting in identification of five off-target sites associated with SpCas9-gMH treatment that previously went undetected by conventional targeted amplicon sequencing[28]. These results showcase the need for highly sensitive off-target detection methods beyond the

limits of regular amplicon sequencing, and for their implementation into guidelines for the development of genetic medicines[12].

In order to minimize off-target activity and thereby mitigate these safety-related risks of therapeutic gene editing, SpCas9 has been subjected to various structure-based engineering or directed evolution-based approaches[61,67–71]. Although some of these engineered SpCas9 variants have fewer off-targets, increased specificity often comes at a cost of a decreased on-target efficiency[72]. Given these limitations encountered by SpCas9 engineering efforts, there is a growing interest in identifying alternative Cas9 enzymes with favorable specificity, smaller size, and broader PAM profile by mining bacterial genomes. These efforts led to the identification of orthologs from various families[73–76] including Type II-B *Francisella novicida* Cas9 (FnCas9)[73–76] and type V Cas12a[36] that display promising specificity. In this work, we mined the human microbiome for additional Cas nucleases, and identified an additional Type II-B Cas9 ortholog. The enzyme described here, PsCas9, discriminates DNA target sites from similar sequences elsewhere in the genome at high precision while retaining its intended activity. We speculate that this specific feature is likely due to the slower kinetics of off-target DNA cleavage by PsCas9 as compared to SpCas9. The most illustrative example of superior off-target discrimination of PsCas9 are our in vivo editing experiments where PsCas9 was able to distinguish between human *PCSK9* and mouse *Pcsk9* sites despite the minimal difference (a single nucleotide between the two genes at position 11 in the protospacer sequence). The presence of the G-U mismatch compromised the activity of PsCas9 by nearly 40% with no impact on SpCas9, highlighting the ability of PsCas9 to efficiently discriminate even highly similar off-target sites from its intended target. The remaining activity of PsCas9 at the human site is likely due to the nature of the mismatch, as U can pair with G as a wobble base[77]. PsCas9 also showed drastically reduced number of hits compared to SpCas9 in CHANGE-seq, an unbiased biochemical assay to experimentally nominate potential genome-wide off-targets. It is important to note that no unique off-targets that were not already detected by SpCas9 were present in PsCas9 samples further highlighting its high specificity.

Consistent with this unique intrinsic fidelity, Duplex-Seq, with its improved sensitivity, did not detect any off-target editing at any of the analyzed sites in PsCas9-treated mice. Genomic modifications catalyzed by PsCas9 showed additional beneficial safety characteristics: chromosomal translocations between the mouse and human *PCSK9* loci upon DSB repair were dramatically reduced with PsCas9 compared to SpCas9. We hypothesize that this is due to the combination of the much slower DNA cleavage kinetics at off-targets and the formation of 3-bp 5'-overhangs upon DNA cleavage by PsCas9, as opposed to the blunt or 1-bp overhangs generated by SpCas9. In line with this hypothesis, a recent study reported that a combination of a sticky cutter nuclease with SpCas9 leads to reduced chromosomal translocations[78].

While PsCas9 outperformed wild-type SpCas9 in terms of editing specificity, it showed similar activity on some sites (*HEK3*, *HEK4*, *CD34*, *HBEGF*, *STAT1 AAVS1*) and compromised editing efficiency on other sites (*FANCF* and *EMX1*). Whether the effect is due to low efficiency of the enzyme itself or high efficiency of DNA repair processing DNA overhangs at those particular sites remains to be studied. Therefore, when applying PsCas9 to inactivate genes, we advise screening for efficient sgRNAs with particular focus on those that do not favor in-frame mutations.

In conclusion, our work sheds light on the importance of improving sensitivity of sequencing methods to enable detection of low-frequency variants that may occur due to off-target activity of genetic medicines. The Duplex-Seq workflow we described here represents one such strategy; however, more work is needed to implement this comparably new technique in standard gene editing

pipelines to harness its full potential in the development of genetic medicines. It should also be noted that the applied Duplex-Seq off-target analysis strategy, despite the improved sequencing sensitivity, retains the limitation of relying on a previously determined set of candidate sites, meaning that off-target mutations not included in this set will be missed. Nonetheless, Duplex-Seq revealed potential liabilities of SpCas9, that prompted us to look for new Cas nucleases, resulting in the discovery and characterization of PsCas9. Altogether, PsCas9 nuclease represents an important genome editing tool and a potentially safer alternative to wild-type SpCas9 enzyme for clinical applications.

## Methods

### Ethical statement
All mouse experiments were approved by the AstraZeneca internal committee for animal studies and the Gothenburg Ethics Committee for Experimental Animals (license numbers: 162-2015+ and 2194-2019) compliant with EU directives on the protection of animals used for scientific purpose.

### In vivo gene editing
For assessment of editing efficiency and off-target activity in vivo, 5 months old male hPCSK9-KI mice[29] were injected with $1.4 \times 10^9$ ifu of adenovirus encoding either for SpCas9 + gMH guide or PsCas9 + gMH guide. Note that gMH expression was driven from the U6 promoter, resulting in an additional G nucleotide at the 5' end of the sgRNA. Adenovirus vectors were produced by Vector Biolabs (Malvern)[28]. A 22 mer spacer sequence was used for PsCas9. One week post-injection, the animals were terminated, and DNA was extracted from liver tissue with the Gentra Puregene tissue kit (Qiagen) according to manufacturer's instructions. On-target editing was assessed by targeted amplicon sequencing followed by CRISPResso2 analysis. Note that SpCas9-GFP mice were injected independently.

### Amplicon sequencing preparation
For cell-based experiments, genomic DNA was extracted 72 h after transfection by QuickExtract DNA extraction solution (Lucigen) according to manufacturer's protocol. Amplicons were generated by using Phusion Flash High-Fidelity Mastermix (Thermo Scientific). Briefly, 0.5 µM of target specific primers with NGS adapters and 1.5 µl of genomic DNA extract was used in a total 15 µl reaction volume. The following cycling conditions were used: 98 °C 3 min, (98 °C 10 s, annealing temperature for each primer pair for 5 s and 72 °C 5 s) for 30 cycles. PCR products were purified by Ampure XP beads (Beckman Coulter). PCR product size and DNA concentration was analyzed on a Fragment Analyzer (Agilent). Next, a second round of PCR was performed to add unique Illumina indexes by KAPA HiFi HotStart Ready Mix (Roche). For this step, 1 ng of template and 0.5 µM of indexing primers were used in a total 25 µl reaction volume. The following cycling conditions were used: 72 °C 3 min, 98 °C 30 s, (98 °C 10 s, 63 °C 30 s and 72 °C 3 min) for 10 cycles and 72 °C 5 min final extension. Indexed amplicons were purified by Ampure XP beads and analyzed by Fragment Analyzer. Amplicons were pooled and quantified by Qubit 4 Fluorometer (Life Technologies). Sequencing was performed using the Illumina NextSeq system according to manufacturer's protocol. For off-target analysis of in vitro experiments and the evaluation of in vivo on-target editing, amplicons were generated by Q5 Hot Start High-Fidelity 2× Master Mix (NEB) according to the manufacturer's protocol. The following cycling conditions were used for off-target analysis: 98 °C 3 min, (98 °C 10 s, 65 °C 15 s and 72 °C 15 s) for 30 cycles and 72 °C 2 min final extension. Amplicon sequencing-based in vivo off-target analysis was done by multiplexed rhAmpSeq™ using a customized IDT rhAmpSeq™ panel and the rhAmpSeq™ Library Kit (IDT) according to manufacturer's instructions. Sequencing was performed on the Illumina NextSeq system according to manufacturer's protocol. Primers

for amplicon sequencing and annealing temperatures are provided in Supplementary Data 2.

## Duplex sequencing

In vivo off-target analysis was performed by Duplex Sequencing with a custom capture panel covering the two on-target sites, 75 off-target sites, and 37 control regions. Genomic libraries were generated from sheared genomic DNA with the xGen Prism Library Preparation kit (IDT) following standard protocol. Target capture was done by two successive rounds of hybridization capture with the xGen Hybridization and Wash kit (IDT), the custom capture panel, TS Mix xGen Universal Blockers (IDT), and Mouse Cot-1 DNA. Library sequencing was performed on a NovaSeq 6000 instrument using an S2 flowcell. Sequencing data was processed with bcbio-nextgen variant calling pipeline v1.2.7 (https://doi.org/10.5281/zenodo.3564938) utilizing a custom genome reference based on mm10 genome with the addition of the PCSK9 transgene content. Reads were aligned to the reference with BWA-MEM (arXiv:1303.3997). Duplex UMI processing, read grouping and collapsing were performed with fgbio (fulcrumgenomics.github.io/fgbio). Occurrence of off-target mutations was analyzed with CRISPResso2[33]. In detail, each Duplex-Seq bait was padded with 100 bp on both ends, intervals were merged and then split into 40 bp-windows for CRISPResso2 WGS analysis to get an overview of mutations. Subsequently, a second CRISPResso2 run was initiated using 20–40 bp sliding windows around each area of interest (i.e., cut site of potential off-targets). To account for the observed differences in cleaving position of SpCas9 and PsCas9, CRISPResso2 analysis for PsCas9-treated samples was done with a cut site parameter of "-w −6" while the "-w −3" standard was used for SpCas9. More detailed description of Duplex-Seq and translocation analysis is provided in Supplementary Data 5. Sequence information of the enrichment probes and the regions that were analyzed during variant calling analysis are detailed in Supplementary Data 6. All Duplex-Seq experiments were performed in parallel in the same sequencing run.

## Droplet digital PCR

For the detection of in vivo translocations, genomic DNA was isolated from mouse liver following the recommendations in the Gentra Puregene Tissue Kit (Qiagen). The DNA concentrations were measured using a Nanodrop and solutions of 10 ng/μl were prepared. Custom FAM-labeled ddPCR-assays were designed to target the following regions: the human hPCSK9 cDNA spanning exon 1 and 2, the endogenous mouse PCSK9 gene, and the translocation event where the 3′ end of mouse exon 3 is fused to the 5′ end of human exon 3. In transfected HEK293T, balanced translocations between either HIST1H2BC-HBEGF or PCSK9-HBEGF were detected using custom FAM-labeled ddPCR assays (BioRad). HEX-labeled AP3B1 assay (BioRad, dHsaCP1000001) was used as reference. Sequences for the primers and probes are listed in Supplementary Data 2.

In an effort to obtain an as accurate result as possible, the primers and probes used in the ddPCR assay were designed to bind in an area at least 50 bp upstream and downstream of the cut site/nickase site (thereby avoiding placing them close to the cut site where potential indels could have prevented the primers or probe from binding, resulting in an underestimated translocation frequency).

A Mastermix was prepared using a final concentration of 1× ddPCR Supermix for Probes, no dUPT (BioRad), 1× FAM-labeled targeting assay (see above), and 1× AP3B1-HEX labeled reference assay 1/40 HindIII (ThermoScientific). 20 μl Mastermix per well to be analyzed was prepared in ultrapure RNase and DNase free water (Invitrogen). After adding 5 μl DNA at 10 ng/μl to the 20 μl Mastermix in the semi-skirted 96-well plate followed by careful mixing, the plate was sealed using a PX1™ PCR Plate Sealer (Bio-Rad). An automated Droplet Generator (BioRAD) was used to generate droplets in a new semi-skirted 96-well PCR plate. The PCR plate was sealed using a PX1™ PCR Plate Sealer

(Bio-Rad). After sealing, the PCR plate was placed in a C1000 Touch™ Thermal Cycler (Bio-Rad) for PCR amplification. The following cycling conditions were used: 95 °C 10 min, (94 °C 30 s, 62–63.2 °C 1 min (primer-specific as indicated) and 72 °C 1 min) for 40 cycles and 98 °C 10 min. The droplet reading was performed with the QX 200 Droplet reader (Bio-Rad) using ddPCR™ Droplet Reader Oil (Bio-Rad).

Data acquisition and analysis was performed using the software QuantaSoft (Bio-Rad). The fluorescence amplitude threshold was set manually as the midpoint between the average fluorescence amplitude of the four droplet clusters (FAM-positive, HEX-positive, positive for both FAM and HEX, and empty droplets) and the negative control. The same threshold was applied to all wells of the ddPCR-plate using the same interrogated assay.

## Bioinformatic methods to identify PsCas9 from the human gut microbiome

To identify Cas9 II-B candidates from the human gut microbiome, a search was performed using hmmscan (version 3.1b2) from the HMMER package[79] using the profile TIGR03031 from the TIGRFAM collection of Hidden Markov Models[80] as the query. The TIGR03031 hmm model, designated as IPR013492 in the Interpro database, includes proteins of CRISPR system Type II-B RNA-guided endonucleases. The set of proteins searched was from the catalog of microbial genes in the integrated genome catalog (IGC) from the human gut microbiome[40] which contains 9.8 million genes. The search resulted in 10 hits with a cut-off above the trusted cut-off (−cut_tc) out of 9.8 million proteins. The contigs from the metagenomic assembly were obtained for the hits and CRISPRs were identified with pilercr[43]. To determine the taxonomic origin of the *cas9* gene produce, BLAST searches were performed. Using blastp with the protein sequence as query and NCBI as database, resulted in the identification of a very high homology to WP_258333620.1 (99%) from *Parasutterella secunda*.

## Cas9 IIB protein sequence tree

Sequence IDs of Type II-A, II-B and II-C were obtained from Makarova et al.[42] and Gasiunas et al.[53]. Obtained protein sequences together with the protein sequence of PsCas9 were aligned with ClustalO (version 1.2.4)[81], a tree was constructed from the multiple sequence alignment with FastTree (version 2.1.10)[82].

## In silico prediction of tracrRNA

Identification of tracrRNA was performed by screening additional repeats on both strands that did not belong to the CRISPR repeat array by allowing 15 mismatches as ref. [41]. The alignments (crRNA:tracrRNA) were analyzed for the presence of conserved structure motifs using the CLC Genomics Workbench to predict RNA secondary structure. Sequences are provided in Supplementary Data 1.

## Purification of Cas9 proteins

Proteins were produced in *E. coli* BL21 λDE3 Star using the T7 expression system and all plasmids were based on pET24a. Freshly transformed cells were grown overnight in LB media, before inoculating with 800 ml of TB media the following day. Expression cultures were then grown at 37 °C with vigorous shaking until $OD_{600}$ ~ 2. The temperature was then lowered to 18 °C and IPTG was added after 1 h to a final concentration of 200 μM. The cells were then left at 18 °C overnight and harvested by centrifugation the following day. Cells were resuspended in 20 mM HEPES pH 7.5, 150 mM KCl, 5% glycerol and 1 mM DTT. Lysis was performed by high pressure disintegration and clarified by centrifugation at 15,000 × g for 30 min. The supernatant was then applied to a 5 ml HisTrap column (Cytiva), and washed with a buffer consisting of: 20 mM HEPES pH 7.5, 150 mM KCl, 5% glycerol, 20 mM imidazole, and 1 mM DTT. Bound proteins were then eluted by washing with the same buffer supplemented with 300 mM imidazole. Protein containing fractions were then applied to a Superdex 200 10/

600 column (Cytiva), equilibrated with 20 mM HEPES pH 7.5, 300 mM NaCl, 5% glycerol, and 1 mM DTT. The protein peak was then collected and concentrated to 10 mg/ml and flash frozen in small aliquots and stored at −80 °C until use. Sequences are provided Supplementary Data 3.

## Identification of PAM

The sgRNA duplex was prepared by annealing 1:1 tracrRNA (Synthego) and crRNA (Synthego) in Duplex Buffer (IDT) by applying following cycling conditions: 95 °C 10 min, then ramp down to 25 °C at 0.3 °C/min. Cas9 and pre-annealed sgRNA duplex were incubated 1:1 in NEB buffer 3.1 at 37 °C for 1 h. 4 N randomized PAM plasmid library (1 μg) and 5 nM Cas9/guideRNA complex were incubated for 1 hr at 37 °C in a 100 μl reaction volume[44]. After 1 h, 20 μl of 10 mg/ml Proteinase K (Sigma) was added to the reaction mixture and incubated for 10 min at 65 °C. The reaction mixture was then purified by Ampure XP beads (Beckman Coulter) and eluted in EB buffer (Qiagen). To create blunt ends, purified linear DNA fragment was incubated with 1U T4 DNA polymerase (Thermo Fisher) and 2 mM of dNTP mix at 11 °C for 20 min and the reaction was stopped by incubation at 75 °C for 10 min. Final product was then purified by Ampure XP beads. To add 3′-dA, purified product was incubated with 1.25U Dream Taq polymerase (Thermo Fisher) and 0.2 mM dATP at 72 °C for 30 min followed by bead purification. Next, 3′-dT overhang containing adapters (10 ng/ul) were prepared by annealing Adapter_forward and phosphorylated Adapter_reverse in Duplex Buffer (IDT) with the following cycling conditions: 95 °C 10 min, then ramp down to 25 °C at 0.3 °C/min. Next, 100 ng of annealed adapter was ligated to 100 ng of 3′-dA overhang containing products with T4 DNA ligase (Thermo Fisher) by incubating for 1 h at room temperature. Ligated product was purified by bead wash and quantified using Qubit 4 Fluorometer (Life Technologies). PCR amplification was performed to enrich ligated products (20 ng) with primers (Enrich_forward and Enrich_reverse) and Phusion High-Fidelity PCR Master Mix (Thermo Fisher) according to manufacturer's protocol and the following cycling conditions were used: 98 °C 30 s, 98 °C 15 s and 72 °C 30 s for 15 cycles, 72 °C 5 min final extension. Enriched product was purified by bead wash and final PCR amplification was performed with primers containing adapters for amplicon sequencing (NGS_forward and NGS_reverse) with the same cycling conditions applied as in the enrichment PCR step. To amplify the uncleaved control library, ControlNGS_forward and ControlNGS_reverse was used with cycling conditions: 95 °C 5 min, 98 °C 1 s, 60 °C 5 s and 72 °C 30 s for 29 cycles, 72 °C 1 min final extension. For preparation of sequencing see section Amplicon sequencing. Sequences are provided in Supplementary Data 2.

## Analysis of PAM identification assay and cleavage site

The resulting sequencing data was filtered by selecting the reads with 12 nt match flanking either side of the 4 N randomized PAM sequence. To compensate for library bias, the frequency of each PAM sequence was normalized to the frequency in the initial library. Position frequency matrix (PFM) was used to calculate the probability of finding a nucleotide in each position and results were visualized by R script (ggseqlogo)[44].

To find the most pre-dominant cleavage position, normalized sequencing data was filtered again based on the reads 11 nt match flanking downstream and 11 nt match flanking from 7 nt upstream of the randomized PAM sequence. Next, the remaining sequences in between were grouped according to the number of bases (reflecting blunt/1-nt/2-nt/3-nt/4-nt overhang) and percentages were visualized as a heat map.

## In vitro cleavage assay

In vitro cleavage was performed by mixing pre-annealed tracrRNA and crRNA (1:1) duplex with Cas9 at 1:1 molar ratio followed by incubation at 37 °C for 1 h in in NEB buffer 3.1. Next, 250 ng (3 nM) of plasmid library (3 N or 4 N) was cleaved with 3 nM, 30 nM and 300 nM RNP complex at 37 °C for 2 h in 30 μl reaction volume. After incubation, RNase A (Qiagen) was added, and the reaction mix was incubated for 10 min at 37 °C. To release the DNA from Cas9, 3 μl of 10 mg/ml Proteinase K (Thermo Fisher) was added to the reaction mixture and incubated for 10 min at 65 °C. Sample buffer (5×, Thermo Fisher) was added to each reaction and products were run on a Novex 10% TBE gel (Thermo Fisher) for 45 min at 180 V. After the run, the gel was removed from the cassette and stained with EtBr (PanReac AppliChem) for 20 min. The gel was imaged by Gel Doc system (Bio-Rad). Sequences of tracrRNA and crRNA is provided in Supplementary Data 2.

## In vitro kinetic characterization

The experimental procedures were adapted from Gong et al.[51]. Single guide RNAs (sgRNA) for SpCas9 and PsCas9 were synthesized by Synthego. RNAs were diluted to 10 μM in Annealing buffer (10 mM Tris-HCl, pH 8.0, 50 mM NaCl, 1 mM EDTA) and refolded by following cycling conditions: 95 °C for 5 min and cooling down to room temperature. Cas9-sgRNA complex was formed via mixing 1 μM purified protein with 2 μM refolded RNA in 1× Cleavage buffer (20 mM TRIS-HCl pH7.5; KCL 100 mM; 5% glycerol; 1 mM DTT; 10 mM MgCl$_2$) and incubation at 37 °C for 10 minutes. The reaction was started by quickly mixing the complex with fluorescently labeled substrate DNA at 100 μM and 10 μM, respectively, at 37 °C. The reactions also contained 10 μM loading control DNA having no PAM or target sequence. Aliquots were taken at various time points and quenched with 0.5 M EDTA to the final concentration of 50 mM. Quenched aliquots were deproteinated with Proteinase K (Sigma). Next, cleavage products were resolved with Capillary Electrophoresis (CE) using ABI 3730xl in 90% Hi-Di™ Formamide (Thermo Fisher). Cleavage kinetics was fitted with double exponential functions using R script and Fragman package[51,83]. Total digested DNA fraction was also estimated at the end of each time course experiment. Fluorescently labeled DNA substrates (Fig. 2E and Supplementary Fig. 4D) were synthesized and labeled with 5′-HEX at non-targeted strand by IDT. Substrates for mismatch experiment (Fig. 2G) were generated using PCR with 5′−6-FAM labeled oligos manufactured by IDT. Sequences are provided in Supplementary Data 2.

## Plasmids

Construction of a plasmid library with randomized PAM sequence was initiated by a PCR reaction with primers (Randomized PAM library_primer1 and primer2) to generate a short amplicon containing either 3 N (NNN) or 4 N (NNNN) randomized barcodes with upstream targeting sequence. Amplified fragment was further digested by XhoI and XbaI according to standard protocol by NEB. Fragments were further purified by QIAquick PCR Purification Kit (Qiagen) according to manufacturer's protocol. Plasmid with a PUC57 backbone was digested with SalI and XbaI according to standard protocol by NEB. After the restriction digestion, 1% Agarose gel was run, and linear fragment was cut out. Linear plasmid backbone was further purified by QIAquick Gel Extraction Kit (Qiagen). Ligation of the short amplicons with either 3 N or 4 N randomized PAM sequences was performed by T4 DNA ligase (NEB) according to standard protocol.

For mammalian expression of PsCas9, FnCas9 and SpCas9, bpNLS-Cas9-1XSV40-3XFLAG-T2A-GFP was synthesized by GenScript and cloned into pcDNA™3.1 (+) by standard restriction enzyme-based cloning. PsCas9 was codon optimized by GenScript. The plasmid vector encoding sgRNAs for SpCas9 were cloned by primer pairs containing overhangs (5′-AAAC- and 5′-ACCG) which were annealed and ligated with AarI digested template using T4 ligase (New England Biolabs)[45]. The plasmid vectors encoding sgRNAs for PsCas9 and FnCas9 were generated by ligating annealed primers with overhangs (5′-AAAC- and 5′-ACCG) into BsaI digested standard vector containing

U6 promoter, spacer sequence and tracrRNA scaffold generated by Gibson Assembly (NEB). Scaffold sequence of for FnCas9 was adapted from Chen et al.[34]. The following cycling conditions were used for annealing primers: 95 °C 10 min, then ramp down to 25 °C at 0.3 °C/min. Plasmid sequences are provided in Supplementary Data 3 and all primer pairs for sgRNA cloning are provided in Supplementary Data 2.

### Cell culture and transfections

HEK293T (GenHunter Corporation, Q401; confirmed using STR profiling with >80% identity matched) cells were maintained in Dulbecco's modified Eagle's medium (DMEM) supplemented with 10% fetal bovine serum (FBS). Cells were cultured in 37 °C with 5% $CO_2$. Cell lines were authenticated and tested negative for mycoplasma.

For transfections, $0.25 \times 10^5$ or $0.3 \times 10^6$ cells were seeded into 96-well or 6-well plates the day before. Cells were transfected with 80 ng of Cas9 plasmid and 36 ng-40 ng sgRNA plasmid. For Western Blot, 100 ng Cas9/100 ng sgRNA plasmid were transfected into one well in 6-well plates. Transfections were performed with FuGENE HD transfection reagent (Promega) with 6:1 ratio according to manufacturer's protocol. For the DNA-PKi experiments, M9831 (available at MedChemExpress), was added to the growth medium 3 h prior transfection with a final concentration of 1 μM. For the knock-in experiments, double-stranded DNA (dsDNA) was prepared by annealing primers in Duplex Buffer (IDT) with a final concentration of 2 μM. The following cycling conditions were used: 95 °C 10 min, then ramp down to 25 °C at 0.3 °C/min. After annealing, 0.4 pmol of dsDNA was added to the reaction mixture together with Cas9/sgRNA plasmids. Sequences of dsDNA donors (blunt and with overhangs) are provided in Supplementary Data 2.

### CHANGE-seq

CHANGE-seq was performed as previously described by Lazzarotto et al.[62] with minimal modifications on human high molecular weight (HMW) genomic DNA (Promega). CHANGE-seq was optimized for SpCas9, which generates predominantly blunt ends. We modified the method for PsCas9 by introducing an additional end repair step for blunting the staggered ends generated upon PsCas9 cleavage in vitro, thus enabling subsequent adapter ligation and sequencing. As expected, the modified end-repair CHANGE-seq detected the PsCas9 on-target cleavage much more effectively than the original CHANGE-seq (average of three replicates: 606 and 54 read counts, 40 and 5 when scaled to per million mapped reads, respectively), at a similar detection efficacy to SpCas9 (average of three replicates: 557 read counts, 50 when scaled to per million mapped reads), and therefore was selected as the method of choice for PsCas9 in subsequent analyses (Fig. 3F and Supplementary Data 7).

Size of genomic DNA was assessed in the Fragment Analyzer (Agilent), and further subjected to tagmentation with a custom Tn5-transposome harboring oCRL225/oCRL226 adapters and the Hyperactive Tn5 Transposase (Diagenode, Cat. No. C01070010-20). DNA tagmentation was performed in batches of 2 μg, utilizing 8.7 μl of the assembled transposome in a final volume of 200 μl of 1× Tagmentation Buffer (Diagenode, Cat. No. C01019042) and incubated for 7 min at 55 °C. Reaction was quenched by the addition of 200 μl of SDS 0.4%, and resultant fragments were assessed on the Fragment analyzer and quantified by Qubit dsDNA BR Assay kit (ThermoFisher). Tagmented DNA was then subjected to gap repair with Kapa Hi-Fi HotStart Uracil+ DNA Polymerase (KAPA Biosystems) and Taq DNA Ligase (NEB). Resultant gap-repaired DNA was treated with USER enzyme (NEB) and T4 polynucleotide kinase (NEB), and then circularized overnight with T4 DNA Ligase (NEB) and treated with a cocktail of exonucleases containing Plasmid-Safe ATP-dependent DNase (Lucigen), Lambda exonuclease (NEB) and Exonuclease I (NEB) to degrade residual linear DNA carryover. 150 ng of circularized material were in-vitro cleaved by

SpCas9 or *wild-type* PsCas9 RNPs in combination with HEK4 sgRNAs (sequences details in Supplementary Data 2) in a total volume of 50 μl. PsCas9 libraries were additionally subjected to an end-repair step using T4 DNA polymerase (NEB), so that 5′ overhangs are filled to form blunt ends for ligation. Then, Illumina Universal Adaptor (NEB) was ligated to adenylated blunt ends, enzymatically treated with USER enzyme (NEB) and amplified with NEBNext Multiplex Oligos for Illumina for 20 amplification cycles. The quality of the amplified and bead-cleaned-up libraries was determined using a 5300 Fragment analyzer with the standard sensitivity NGS kit (Agilent). Libraries were further quantified by qPCR (ThermoFisher), pooled and denatured according to Illumina's recommendations and sequenced on a NextSeq550 on a PE150 configuration, to achieve a mean coverage of 5 M reads per library.

The sequenced data was analyzed using the previously published CHANGE-seq analysis pipeline (https://github.com/tsailabSJ/changeseq) with minor modifications. The pipeline was run with the following parameters: read_threshold: 4, window_size: 3, mapq_threshold: 50, start_threshold: 1, gap_threshold: 3, mismatch_threshold: 6, search_radius: 30, merged_analysis: False, target sequence: GGCACTGCGGCTGGAGGTGGNGG. Reads with MAPQ = 0 were included in the analysis alongside those passing the MAPQ threshold defined in the parameters, in order to nominate putative off-targets located in non-uniquely mappable regions.

### Western blot

After 48 h of transfection, cells were washed with cold PBS and incubated on ice for 30 min with 200 μl of lysis buffer (50 mM Tris-HCl, pH 7.4, 150 mM NaCl, 1 mM EDTA, 1% vol/vol Triton X-100) supplemented with EDTA-free Protease Inhibitor Cocktail (Merck). Following incubation, supernatant was taken after spinning down for 10 min at $20,000 \times g$. Cell lysates were preheated for 15 min at 70 °C with NuPAGE LDS Sample Buffer (4×) (Thermo Fisher) containing 1 mM Dithiothreitol (DTT). Proteins were separated on NuPAGE 4–12% Bis-Tris gel (Invitrogen) and transferred onto Nitrocellulose membrane (LICOR Odyssey). The membranes were then blocked with LI-COR Blocking Buffer (LI-COR Biosciences GmbH) and probed with specific primary antibodies. Western blot signals were scanned by using Odyssey Imager from LI-COR Biosciences GmbH. The following primary antibodies were used for detection: anti-Flag (F1804, Sigma-Aldrich; dilution 1:2000) and anti-GAPDH as an internal control (D16H11, CST; dilution 1:10,000). The following secondary antibodies were used for detection (all 1:10,000 dilution): goat anti-mouse 800CW, goat anti-rabbit 800CW, goat anti-mouse 680LT, or goat anti-rabbit 680 (all from LI-COR Biosciences GmbH).

### Bioinformatic analysis

NGS data was demultiplexed by using bcl2fastq software. The fastq files were analyzed by CRISPResso version 2.1.1[33]. For NGS data analysis of experiments performed in HEK293T cells, following parameters were used: -q 30 –ignore substitutions max_paired_end_reads_overlap 300 - w 5. The quantification_window_center of "wc −6" was used for PsCas9 and FnCas9 while the "wc −3" standard was used for SpCas9. For the knock-in analysis, reads were filtered based on containing the knock-in sequence either in forward or reverse direction within the total analyzed reads and reported as a frequency of knock-in. For sgRNA designs, AstraZeneca proprietary software was used as an in-silico tool, which is based on Wellcome Trust Sanger Institute's codebase (WGE: http://www.sanger.ac.uk/htgt/wge/). STAT1 and CD34 sgRNAs were adapted from Overbeek et al. and sgRNA for HBEGF was adapted from Li et al.[64]. For in vivo editing, sgRNAs are described in Akcakaya et al.[28]. Detailed parameters for CRISPResso2 analysis are provided in Supplementary Data 4. Analysis of mutation pattern in Supplementary Fig. 5B was performed as described in Taheri-Ghahfarokhi et al.[45].

## Pcsk9/PCSK9 protein assessment in plasma

Over the course of the study, mouse blood was collected into EDTA-coated tubes from the vena saphena (pre-injection) and from cardiac puncture (during termination). Tubes were kept on ice and plasma was isolated by 20 min centrifugation at $12,000 \times g$ and 4 °C. For assessment of mouse Pcsk9 and human PCSK9 plasma levels, the samples were diluted 1:1000 and 1:800, respectively, and measured with standard ELISA kits (DPC900 and MPC900; R&D Systems, Minneapolis, MN, USA) according to the manufacturer's instructions.

## Data analysis and presentation

Data are presented as mean ± SD if not indicated otherwise. Analysis was performed with Prism 6 (GraphPad Software). Statistical significance was determined using either a two-way ANOVA followed by Sidak's multiple comparison test or an unpaired two-tailed $t$ test, as appropriate for each experimental set-up. $P \le 0.05$ was considered statistically significant. Figures 1A and 4A were created with BioRender.com.

## Statistics and reproducibility

Sample sizes were determined based on literature precedence for genome editing experiments. For the animal experiments, no sample size calculation was performed. The sample size was determined as the generation of triple-independent samples for comparisons between groups that is sufficient to perform statistical tests. All cell data and in vitro experiments were independently repeated at least once as specified in figure legends. Mammalian cells were cultured under identical conditions, no randomization or blinding was used. Animals were randomized based on their weights measured prior to the experiments and no blinding was applied.

## Reporting summary

Further information on research design is available in the Nature Portfolio Reporting Summary linked to this article.

# Data availability

Next-generation sequencing data from amplicon sequencing and CHANGE-seq are available in the NCBI Sequence Read Archive database (SRA) under BioProject accession code PRJNA1000737 [https://www.ncbi.nlm.nih.gov/bioproject/1000737]. The experimental data generated in this study are provided in the Source data file. Source data are provided with this paper.

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

## Acknowledgements

We thank Steve Rees for supporting this work and members of the Genome Editing Department for suggestions and comments on the manuscripts. We thank Marie Johansson, Liselotte Andersson, and Sara Torstensson for the help with the in vivo work. We thank Damla Etal, Maryam Clausen, Inken Dillmann, and the NGS facility for their support. We thank Life Science Editors for language correction. This project has received funding from the European Union's Horizon 2020 research and innovation program under the Marie Skłodowska-Curie grant agreement no. 765269 (S.W.). B.B. is a previous Postdoctoral Fellow of the Astra-Zeneca R&D Postdoc program. A.M. and A.R. are Postdoctoral Fellows currently in the AstraZeneca R&D Postdoc program.

## Author contributions

B.B., A.T.-G. and M.M. initiated the project. B.B., S.W., D.D., and A.M. performed most of the experimental work with help from A.R., S.C., N.S., J.L.T., M.F., E.E., E.G., S.L., L.B., C.I.M., A.T.-G., P.A., and G.S. F.K. and M.F. performed bioinformatic analysis. A.M., S.N., M.C., B.N., and P.A. performed Duplex-Seq and its analysis. K.C., M.B.-Y., M.S., and M.P. provided technical input and guidance. B.B., A.M., A.R., and G.S. prepared the manuscript with input from all authors. G.S. and M.M. supervised the study.

## Competing interests

S.W., D.D., A.M., A.R., F.K., J.L.T., M.F., S.C., N.S., E.E., E.G., M.F., M.B.-Y., M.S., M.P., P.A., G.S., and M.M. are employees and shareholders of AstraZeneca. B.B., C.I.M., L.B., S.L. and A.T.-G. are former employees of AstraZeneca. M.M., A.T.-G., F.K., M.B., are co-authors on patent application (PCT/US2018/061680). The other authors declare no competing interests.
