## [Peer Review File · Nature Communications]

Reviewers' Comments:

Reviewer #1:

Remarks to the Author:

This study by Bestas et al. identified a new Cas9 nuclease (named SpOT-ON) designed to reduce the off-target effects compared to SpCas9. The new Cas9 is derived from the II-B subfamily and is proposed to have a higher specificity and to lower chromosomal translocation formation. In parallel, the authors describe a duplex based method to quantify editing at predicted off-target sites with a higher sensitivity compared to amplicon-Seq.

Despite the great interest in describing a new Cas9 variant with a higher specificity, the experiment data do not always fully support the conclusions and some important insights and extensive data are missing and should be addressed by the authors.

Major concerns:

1-For SpotON Cas9 improved specificity:

The Spot-ON Cas9 has a residual 20% editing efficiency on the hPCSK9 locus. This locus has a single mismatch at position 11 compared to the targeted mPCSK9 locus. Even if the editing efficiency is decreased by ~40% compared to spCas9, this mismatch is located between the position 1-16, position where the SpotON activity is supposed to be drastically reduced as shown by the authors (Fig3D). Thus, we could have expected a greater decrease of editing for this off-target locus.

Thus an overall concern is that the authors should give a deeper characterization of the off-target effects. The authors focus their analysis on predicted off target sites expected for the spCas9 for only one locus. As Spot-ON Cas9 is a novel variant, the higher specificity should be assessed through a more genome wide method to make sure that other unpredictable off-target sites are editing and for more than one targeted locus.

In terms of efficiency to cleave targeted sites, SPOT-ON Cas9 appears as less efficient (50% less efficient) for 2 out of 4 loci in Fig3E.

Another major concern is that the SPOT-ON Cas9 induces mostly in frame mutations. This can lead to residual protein expression (as shown in FigS6A) with potential toxic activities .

2- Concerning the Duplex-Seq assay used to detect off-target editing:

One major limiting factor is the design of the probe for all potential off-target sites. Indeed only 75 out of 79 "met technical requirements during probe design". Thus 4 potential off target sites are not evaluated for editing compared to amplicon sequencing.

In addition, the amplicon sequencing for OT-2 and OT-4 (Figure 1C) show respectively 0.04% and 0.08% of editing so these off-target sites were already detected (why these numbers are not in red is unclear). For OT-2, the % of edition is even higher by amplicon seq? (0;08% compared to 0.01%by Duplex-Seq)? In addition, the sensitivity is still low since for 3 of the off-target edited loci (OT-1, OT-3 and OT-5) only 1 of the triplicate show editing >0%.

In consequence, it appears that the Duplex-seq show only moderate improvement in sensitivity to detect off-target sites.

3- Translocation formation after SPOTon Cas9 cleavage:

Overall, more data are needed to formally prove that chromosomal translocation is decreased upon SPotONCas9 cleavage. In addition, the method of detection by ddPCR of translocations is probably not the most suitable to detect these events in mouse cells:

In a mechanistic point of view, there is not rational to support the idea that SPOTon Cas9 will induced less translocations (see below).

The authors should look at chromosomal translocation formation in their in vitro model between several loci (they have 4 edited loci) to demonstrate that SPOTon Cas9 cleavage is less prone to

form translocation.

Concerning the *in vivo* data, the authors did not strictly demonstrate that the translocation formation is reduced with the SPOTon Cas9. Translocation frequency is linked to the cutting efficiency. Since the hPSC9 editing efficiency is decreased by 40% when using SPOTon Cas9, the translocation frequency will be decreased as well, and not only because the SPOTon Cas9 lead to less translocations.

In addition, the quantification of translocations by droplet PCR needs accurate settings and is not very sensitive. It is possible to compare translocation frequency with ddPCR but it does not directly answer the question of the formation of translocation after SPOTon Cas9 cleavage. In the contrary for detection of low translocation frequency, a simple PCR done on DNA serial dilutions is more accurate to detect translocated events at a frequency $< 10^{-6}$.

Furthermore, the ddPCR requires very small amplicons (< 100 bp) which are not suitable for accurate translocation formation detection. Deletions/insertions are induced at breakpoint particularly in mouse cells (which are mostly using altNHEJ to repair translocation breakpoints) (Reviews: Ramsden et al, 2021, DOI: 10.1038/s41388-021-01856-9; Brunet and Jasin, 2018, DOI: 10.1007/978-981-13-0593-1_2). The lack of detection of "modified" breakpoints may artificially decrease the translocation frequency obtained by ddPCR. As described by the authors, SPOTon induced insertions.

In their previous study (Carreras et al, BMC biology 2019), the authors showed high level of formation of the acentric translocated chromosome that is not quantified when using the SPOT ON Cas9.

In the second paragraph the authors wrote the sentence " While the mechanism of CRISPR/Cas9-induced DNA translocations remains largely elusive, the blunt- ended DNA formed after the SpCas9 cut seems to promote this process », this sentence is inaccurate since numerous studies have deciphered in details the molecular mechanism of chromosomal translocations both in human and mouse cells (for reviews: Ramsden et al, 2021, DOI: 10.1038/s41388-021-01856-9; Brunet and Jasin, 2018, DOI: 10.1007/978-981-13-0593-1_2). Similarly, the sentence "its nuclease activity generated 3-nt long 5'-overhangs in the DNA substrate, which suggests that this Cas9 nuclease may decrease the risk of translocations » is misleading: translocations can be easily induced by the use of TALENs and Zinc Finger Nucleases that typically lead to 5'overhangs formation. The authors do not provide any references for their statements...

Other concern: it is unclear what new results are described in the Figure 1 E and F (main Figure) compared to the previous paper of the same authors. In this former paper (Carreras et al, BMC biology 2019), the authors already showed translocation formation from the same exact loci using DNA from the same organs of the same transgenic mice with spCas9.

Minor concerns:

S5: "f" is missing in the legend.

FigS6: no legends on the graph A

In conclusion, the SPOTon Cas9 is a promising nuclease but a deeper characterization of the higher specificity is needed to really demonstrate a significant improvement of the targeting. The induction of in frame mutations may be a serious drawback for the use of this Cas9 for KnockOut experiments for example.

The data concerning the decrease in translocation formation do not fully support the conclusions.

Reviewer #2:

Remarks to the Author:

Bestas et al. performed a Duplex-seq-based candidate off-target sequencing method to reduce sequencing error and to identify low-frequency off-target mutagenesis induced by CRISPR-Cas9 treatment. Next, the authors performed a database search on TIGFRAM and identified a novel MH0245 Cas9 which belongs to type II-B CRISPR. The MH0245 Cas9 cuts DNA slightly slower than SpCas9 and generates a 3-nt long 5'-overhang. Lastly, the authors delivered the MH0245 Cas9 into a mouse model by an adenoviral vector. They identified greater specificity than SpCas9, and

less inter-chromosomal translocation between mouse Pcsk9 on chromosome 4 and the human PCSK9 gene inserted into the Rosa26 locus on chromosome 6. This is a well-written manuscript with properly performed experimental data, and the novel MH0245 Cas9 may have broad applicability once it becomes available to the genome editing community. However, the reviewer has some concerns regarding the manuscript, which are described below.

Major Comments

1) Nomenclature of MH0245 Cas9

Although the authors named the novel type II-B MH0245 Cas9 as "SPOT-ON", this naming would confuse the scientific community, as the authors used the native MH0245 Cas9 amino acid sequence without protein engineering (except for codon optimization and addition of NLS). Like "Cpf1" has been re-named as "Cas12a" to retain the consistency of CRISPR nomenclature, the reviewer strongly recommends maintaining the "Cas9" and adding a prefix of specie origin (or other ID like MH0245) to be consistent with another CRISPR-Cas9s, such as SpCas9 or FnCas9.

P32. "Using blastp with the MH0245 Cas9 protein sequence as query and NCBI nr as database resulted in related proteins sequences from a metagenomic species annotated as Succinatimonas and several Sutterella species. However, the hit with the highest score had only 37% protein sequence identity to MH0245 Cas9. "

When the reviewer performed a blastp search against the non-redundant protein sequences database with a default blastp setting, the Cas9 protein sequence from *Parasutterella secunda* was identified with 99% identity (Sequence ID: WP_258333620.1). Hence, the MH0245 Cas9 can be named "*Parasutterella secunda* Cas9" or "PsCas9" for short, unless there is evidence that the MH0245 Cas9 is not derived from *Parasutterella secunda*.

2) Data presentation on in vitro DNA cleavage activity

Fig. 2F, 2G: In addition to the cleavage kinetics (k_{fast}), please also show the "fraction digested" in the graphs. As shown in Figure S4D or S4G, although the cleavage speed of MH0245 Cas9 was 20-300 seconds slower than SpCas9, the final cutting efficiency reached a similar level at the end. When considering the off-target cleavage risk, absolute digestion efficiency should be indicated.

3) Duplex-seq method

P3. "Furthermore, most methods used for the evaluation of editing outcomes, including "Verification of in vivo off-targets" (VIVO) strategy, are limited in their sensitivity by the current detection limit of targeted deep sequencing of 0.1%."

The "VIVO" method is a combination of in vitro CIRCLE-seq to identify candidate sites and in vivo deep sequencing analysis. The Duplex-seq may be an advancement of the targeted deep sequencing analysis part, but it still relies on the identification of candidate sites. Although Duplex-seq is a powerful method for reducing the noise caused by sequencing errors, the searchable area is limited to the pre-determined candidate sites. These limitations should be mentioned in the Discussion.

Figure S2 and S3

If the sensitivity of Duplex-seq is 0.01%, please indicate the data with two digits after the decimal point (i.e., "0.03%" rather than "0.0%").

4) Cleavage efficiency comparison between MH0245 Cas9 and SpCas9.

P2. "Once loaded with optimal protospacer length, MH0245 cleaves DNA substrates at a similar

rate to SpCas9 (Figure 2G, Supplementary Figure 4J)."

Provided Figure 2G and Supplementary Figure 4J do not have side-by-side comparison data of MH0245 Cas9 and SpCas9. Considering MH0245 Cas9 showed lower genome editing activity in some gRNA targets in HEK293T cells (Fig. 3E), whether the same gRNA sets perform similarly in vitro should be investigated.

5) Translocation efficiency between Fig. 1 and Fig. 4

Although the on-target editing efficiency was greater in Figure 1 than in Figure 4, the translocation efficiency of SpCas9 in Figure 1 (around 0.16%) was lower than that in Figure 4 (about 0.4-0.6%). Why did this discrepancy happen? If this is due to the artifact of deep sequencing, it is recommended to measure the translocation efficiency of Figure 4 samples by ddPCR, similar to Figure 1.

Minor comments

"CRISPR/Cas9" should be "CRISPR-Cas9".

"effective binding of the Cas-sgRNA complex to DNA requires a protospacer adjacent motif (PAM) upstream of the targeted DNA sequence."

The term "upstream" here may not be appropriate for SpCas9 or FnCas9. Please revise (downstream or 3' side).

Figure S6A

Please indicate the graph legend for the black box, red circle, and green triangle.

Reviewer #3:

Remarks to the Author:

Bestas et al. discuss their findings relating to a novel CRISPR-Cas9 variant with lower off-target editing and demonstrate it's effective for in vitro and in vivo editing. Furthermore, they present application of an existing sequencing methodology with improved sensitivity, which is important for further exploration of CRISPR technologies in a clinical setting. The manuscript is well written and provides strong argumentation. However, there are a few comments/suggestions that require more explanation.

- Page 4: How are significance levels determined between targeted amplicon sequencing and Duplex-seq? Also, how is significance calculated between treated and untreated for Duplex-seq?
- Page 5: What is the sequence homology between MH0245 and SpCas9?
- Page 7: The authors describe mismatch tolerance for positions 6-13 and 15-18. Are there potential mismatches in the queried off-target sites?
- Page 7: The authors mention a prevalent indel profile of 3nt, which can be explained by the nature of the overhangs. This finding should be verified over a larger number of target sites/guideRNAs.
- Page 8: Can you please describe the dsDNA oligo in more detail? Are there any homologous arms present as the be able to differentiate between NHEJ and HDR? If not, what is the ability of MH0245 to do HDR?
- Page 9: The authors should do off-target nomination studies for their novel Cas9-variant. The lower levels at off-target editing using sites defined by SpCas9 look very promising, but it does not rule out that MH0245 can have a distinct off target profile from SpCas9.
- Page 10: If MH0245 has a unique off target profile, putative translocations will not be found with the proposed approach which relies on the SpCas9 off target profile.

Response to reviewers

Bestas B. et al. - SPOT-ON is a Cas9 nuclease with no detectable off-targets and reduced chromosomal translocations in vivo.

Thank you for reviewing our manuscript for publication in Nature Communications, and for your suggestions regarding how to improve it. We were pleased to learn that the reviewers appreciated our work. Below we outline our response to the comments and detail new results we generated to strengthen our conclusions. The outcome of these experiments is now included in our revised manuscript.

Reviewer 1

Remarks to the Author:

This study by Bestas et al. identified a new Cas9 nuclease (named SpOT-ON) designed to reduce the off-target effects compared to SpCas9. The new Cas9 is derived from the II-B subfamily and is proposed to have a higher specificity and to lower chromosomal translocation formation. In parallel, the authors describe a duplex based method to quantify editing at predicted off-target sites with a higher sensitivity compared to amplicon-Seq. Despite the great interest in describing a new Cas9 variant with a higher specificity, the experiment data do not always fully support the conclusions and some important insights and extensive data are missing and should be addressed by the authors.

Thank you for reviewing the manuscript and for highlighting that our findings are novel and interesting. We agree that further characterization and expansion of our previous experiments would strengthen our findings. To that aim, we have now performed a series of additional experiments (including unbiased off-target detection and more sophisticated translocation studies) and elaborated on previous results to support our conclusions of SpOT-ON as a novel Cas9 nuclease with no detectable off-targets and reduced translocations. Please see our point-by-point responses below.

Major concerns:

1-For SpotON Cas9 improved specificity:

The Spot-ON Cas9 has a residual 20% editing efficiency on the hPCSK9 locus. This locus has a single mismatch at position 11 compared to the targeted mPCSK9 locus. Even if the editing efficiency is decreased by ~40% compared to spCas9, this mismatch is located between the position 1-16, position where the SpotON activity is supposed to be drastically reduced as shown by the authors (Fig3D). Thus, we could have expected a greater decrease of editing for this off-target locus.

We thank the reviewer for the valuable comment. Indeed, in our in vivo model SpOT-ON still shows a significant albeit reduced activity at the *hPCSK9* locus containing a single mismatch to the gMH guide RNA, whereas in cell culture a single mismatch at a similar position resulted in almost complete ablation of the editing activity as shown in Fig3D. Importantly, these two experiments were performed with different sgRNA spacers, targeting different sites, the *PCSK9* and

EMX1a, respectively. It was postulated previously (PMID: 33608277) that not only mismatch position, but also sequence context is determining the off-target activity of Cas enzymes. This was further proven structurally in an elegant study from the Jinek lab (PMID: 36306733). Moreover, the gMH guide and the *hPCSK9* DNA target form a rU:dG mismatch, which is better tolerated due to the wobble base pairing mechanism. This mechanism allows for editing activity of both enzymes, but with reduced effect for SpOT-ON Cas9 due to its observed higher fidelity.

Collectively, the residual activity of SpOT-ON Cas9 at the *hPCSK9* locus in the mouse model might seem surprising, but this finding doesn't contradict the presented data and goes along well with existing views on mismatch tolerance in the field. To improve the manuscript clarity for the reader we have :

1. Clearly stated that *EMX1a* locus was used in Figure 3D.

2. Clarified this original text:

“Importantly, while SpCas9 showed similar on-target efficiency at both human and mice loci despite the single G-U mismatch at position 11, SpOT-ON distinguished both loci and edited the human PCSK9 locus ~40% less than the mouse Pcsk9 locus (Figure 4B).”

Updated text:

“Importantly, while SpCas9 showed similar on-target efficiency at both human and mice loci, SpOT-ON distinguished both loci and edited the human PCSK9 locus at ~40% lower efficiency than the mouse Pcsk9 locus despite the rU-dG mismatch that is often tolerated due to wobble base pairing (Figure 4B).”

3. Clarified this original text:

“This increased sensitivity to mismatches in vivo is in line with our in vitro and HEK293T cells observations.”

Updated text:

“This improved fidelity in vivo is in line with our observations of increased mismatch tolerance in vitro and in HEK293T cells for EMX1a targeting guide RNA. “

Thus an overall concern is that the authors should give a deeper characterization of the off-target effects. The authors focus their analysis on predicted off target sites expected for the spCas9 for only one locus. As Spot-ON Cas9 is a novel variant, the higher specificity should be assessing through a more genome wide method to make sure that other unpredictable off-target sites are editing and for more than one targeted locus.

We agree with the reviewer. To more extensively characterise the off-target properties of this novel enzyme, we performed CHANGE-seq which is a genome-wide off-target discovery approach (PMID: 32541958). We profiled SpCas9 and SpOT-ON-activities using the promiscuous HEK4 sgRNA. Briefly, we identified many off-target cleavage sites for SpCas9 (1087, 655 and 651 sites in the three technical replicates respectively. Among these, 372 sites were shared across all replicates) (Figure 3G).

In contrast, for SpOT-ON, the number of detected off-target cleavage sites were much fewer: 49, 31 and 31 sites in the three technical replicates respectively, with only 6 sites being shared across all replicates (Figure 3G, Supplementary Data Table 7). Those 6 off-target sites were also present in the SpCas9 samples and all of them contained the NGG PAM.

In terms of efficiency to cleave targeted sites, SPOT-ON Cas9 appears as less efficient (50% less efficient) for 2 out of 4 loci in Fig3E.

We thank the reviewer for the comment. For the experiment presented in Figures 3E and 3F initially, we used the same guide RNA spacer of 20-nt (which is suboptimal for SpOT-ON as we showed in Figure 2). We have chosen this experimental setup to compare the off-target properties of the enzymes guided to the same sequences. However, we realize that using 20-nt spacer guide RNA with SpOT-ON Cas9 might reduce off-target editing due to overall lower activity as shown in Figure 2. Thus, we repeated and expanded this experiment and compared the on- and off-target editing of SpOT-ON, SpCas9 and FnCas9 with both 20 and 22-nt spacers. This updated data set will completely substitute panels D and E in Figure 3. We also modified the text in the manuscript accordingly. In the updated data set we have indeed noticed that at some sites the SpOT-ON Cas9 has lower performance than SpCas9 despite being used with optimal spacer. However, at other targets shown in FigS5A the activity of both enzymes is comparable. We concluded and mentioned it in the text that this discrepancy is likely to be sequence and locus specific, which is a known property of Cas9 enzymes.

Another major concern is that the SPOT-ON Cas9 induces mostly in frame mutations. This can lead to residual protein expression (as shown in FigS6A) with potential toxic activities.

SpOT-ON generates DNA overhangs as a result of its catalytic activity. Those overhangs will often, but not always, result in three nucleotides insertions among other outcomes of DNA repair. We have noticed that the effect is site-specific and depends on the DNA sequence context. For example, we very often see MMEJ-type of repair at many loci as shown in the updated and expanded Supplementary Figure 5B. In addition 3 nucleotides insertions could result in gene knock-out when targeting introns-exons junctions. We agree that when using SpOT-ON to inactivate genes, the user needs to screen for sgRNA sequences not favouring in frame mutations.

2- Concerning the Duplex-Seq assay used to detect off-target editing:

One major limiting factor is the design of the probe for all potential off-target sites. Indeed only 75 out of 79 “met technical requirements during probe design”. Thus 4 potential off target sites are not evaluated for editing compared to amplicon sequencing.

We thank the reviewer for this question and would like to clarify that the 79 off-target sites that were analyzed in the original amplicon-sequencing experiment did not reflect all potential off-target sites of the gMH sgRNA. In the original VIVO publication, CIRCLE-seq identified 529 potential off-target sites, of which only a subset was chosen for validation by amplicon-sequencing. The aspiration of the present study was not to provide an extensive interrogation of all possible off-target sites, but rather to compare the performance of the amplicon sequencing and Duplex-sequencing pipelines on a subset of off-target sites. Therefore, we are not concerned about the drop-out of the four mentioned sites.

In addition, the amplicon sequencing for OT-2 and OT-4 (Figure 1C) show respectively 0.04% and 0.08% of editing so these off-target sites were already detected (why these numbers are not in red is unclear). For OT-2, the % of edition is even higher by amplicon seq? (0.08% compared to 0.01% by Duplex-Seq)? In addition, the sensitivity is still low since for 3 of the off-target edited loci (OT-1, OT-3 and OT-5) only 1 of the triplicate show editing >0%.

In consequence, it appears that the Duplex-seq show only moderate improvement in sensitivity to detect off-target sites.

We thank the reviewer for pointing this out. The data shown in Figure 1C is derived by amplicon sequencing which has a lower limit of detection of 0.1%. Editing events that are measured at frequencies below this threshold (i.e. all events measured at frequencies <0.1%) are non-distinguishable from random sequencing error-induced events, and therefore cannot confidently be classified as real editing events. The reported values of 0.04% and 0.08% thus fall below this threshold. Therefore, they have not been colored red and cannot directly be compared to the Duplex-seq-derived data. In contrast to amplicon sequencing, the frequencies obtained by Duplex-seq are above the respective sensitivity threshold of the method (i.e. 0.01%) allowing us to regard these events as actual editing events. The advantage of the 10-fold increase in sensitivity achieved by Duplex-seq is thus the ability to more confidently distinguish real off-target events from sequencing-induced background. The reason why not all detected off-target sites were identified to the same degree in all three animals is likely to be natural variability that is always present in *in vivo* studies. While conditions in *in vitro* experiments are more controllable and reproducible, small physiological differences between individual mice might have an impact on the editing outcome leading to detectable editing in some animals while others might only show below-threshold editing.

3- Translocation formation after SPOTon Cas9 cleavage:

Overall, more data are needed to formally prove that chromosomal translocation is decreased upon SPOTONCas9 cleavage. In addition, the method of detection by ddPCR of translocations is probably not the most suitable to detect these events in mouse cells:

In a mechanistic point of view, there is not rational to support the idea that SPOTon Cas9 will induce less translocations (see below).

The authors should look at chromosomal translocation formation in their *in vitro* model between several loci (they have 4 edited loci) to demonstrate that SPOTon Cas9 cleavage is less prone to form translocation.

Concerning the *in vivo* data, the authors did not strictly demonstrate that the translocation formation is reduced with the SPOTon Cas9. Translocation frequency is linked to the cutting efficiency. Since the hPSC9 editing efficiency is decreased by 40% when using SPOTon Cas9, the translocation frequency will be decreased as well, and not only because the SPOTon Cas9 lead to less translocations.

We thank the reviewer for the suggestion to further investigate translocation rates. To address this concern, we have performed additional experiments in HEK293T cells and

measured the chromosomal translocations by ddPCR. We co-transfected cells with two gRNAs that were previously shown to induce translocations, and used established ddPCR assays (PMID: 33479216, <https://doi.org/10.1101/2022.12.15.520396>) to measure balanced translocation events between *HIST1H2BC-HBEGF* and *PCSK9-HBEGF* loci. We measured the on-target single gRNA editing efficiencies of both SpCas9 and SpOT-ON at all loci with amplicon-seq. Since both variants were similarly efficient at the *HIST1H2BC* and *PCSK9* loci, we used the relative editing efficiency at the HBEGF locus to normalize the raw translocation frequencies measured by ddPCR to account for the discrepancy in cutting efficiency at certain loci mentioned by the reviewer. Even after normalization, we found that SpOT-ON drastically is associated with reduced translocation frequencies between simultaneously cleaved sites and have added these experiments in Figure 3 H-I with a corresponding text in the Results section.

In addition, the quantification of translocations by droplet PCR needs accurate settings and is not very sensitive. It is possible to compare translocation frequency with ddPCR but it does not directly answer the question of the formation of translocation after SPOTOn Cas9 cleavage. In the contrary for detection of low translocation frequency, a simple PCR done on DNA serial dilutions is more accurate to detect translocated events at a frequency < 10⁻⁶.

We would like to clarify that here we aimed at quantifying rather than detecting translocations. Regular PCR would not allow us to compare the frequency at which translocations happen after genome editing executed by SpOT-ON or SpCas9. In contrast to regular PCR, Droplet Digital PCR (ddPCR) is a quantitative and more precise method. Instead of providing an end-point result from a single amplification event for each sample, ddPCR allows ~15 000-20 000 individual amplification reactions to take place per sample, and most of these contain only one template, avoiding any bias from preferential amplification of shorter products seen in regular PCR. Also, it is more precise than regular PCR due to a lower risk of off-target amplification as not just the primers, but also the probe, needs to bind DNA (when using probe-based ddPCR-assays as we did here).

In fact, several publications have described ddPCR as a highly sensitive method, for those by Kojabad et al. PMID 33538349, Campomenosi et al, Hindson et al. PMID 23995387, and RT-ddPCR was identified as more sensitive than qPCR e.g. in Robinson et al, PMID 29506703, and Tsujimoto et al, PMID 36378656.

Furthermore, the ddPCR requires very small amplicons (<100 bp) which are not suitable for accurate translocation formation detection. Deletions/insertions are induced at breakpoint particularly in mouse cells (which are mostly using altNHEJ to repair translocation breakpoints) (Reviews: Ramsden et al, 2021, DOI: 10.1038/s41388-021-01856-9; Brunet and Jasin, 2018, DOI: 10.1007/978-981-13-0593-1_2). The lack of detection of “modified” breakpoints may artificially decrease the translocation frequency obtained by ddPCR. As described by the authors, SPOTon induced insertions.

We thank the reviewer for pointing this out and want to clarify the design of our ddPCR experiments. We carefully designed the ddPCR-assay to ensure we would detect translocations even if indels form around the break points.

From the methods section: *“In an effort to obtain an as accurate result as possible, the primers and probes used in the ddPCR assay were designed to bind in an area at least 50 bp upstream and downstream of the cut site/nickase site (thereby avoiding placing them close to the cut site where potential indels could have prevented the primers or probe from binding, resulting in an underestimated translocation frequency).”*

In their previous study (Carreras et al, BMC biology 2019), the authors showed high level of formation of the acentric translocated chromosome that is not quantified when using the SPOT ON Cas9.

The experimental procedure of the Duplex-Seq set-up in fact allows detection of all four possible combinations of translocations between the mouse and human *PCSK9* loci (i.e. acentric, dicentric, as well as the two balanced translocations). We have added the data for both SpCas9 and SpOT-ON (Supplementary Figure 6C-D). While the absolute number of overall translocations was significantly lower for SpOT-ON vs. SpCas9, both enzymes showed a similar distribution of detected translocations with dicentric and acentric translocations being most common.

In the second paragraph the authors wrote the sentence “ While the mechanism of CRISPR/Cas9-induced DNA translocations remains largely elusive, the blunt- ended DNA formed after the SpCas9 cut seems to promote this process », this sentence is inaccurate since numerous studies have deciphered in details the molecular mechanism of chromosomal translocations both in human and mouse cells (for reviews: Ramsden et al, 2021, DOI: 10.1038/s41388-021-01856-9; Brunet and Jasin, 2018, DOI: 10.1007/978-981-13-0593-1_2). Similarly, the sentence “its nuclease activity generated 3-nt long 5'-overhangs in the DNA substrate, which suggests that this Cas9 nuclease may decrease the risk of translocations » is misleading: translocations can be easily induced by the use of TALENs and Zinc Finger Nucleases that typically lead to 5'overhangs formation. The authors do not provide any references for their statements...

We thank the reviewer for insightful comments and for pointing out critical references that we unfortunately missed.

These misleading sentences have now been removed from the manuscript:

- *“While the mechanism of CRISPR/Cas9-induced DNA translocations remains largely elusive, the blunt-ended DNA formed after the SpCas9 cut seems to promote this process. In fact,...”*
- *“which suggests that this Cas9 nuclease may decrease the risk of translocations”*

Other concern: it is unclear what new results are described in the Figure 1 E and F (main Figure) compared to the previous paper of the same authors. In this former paper (Carreras et al, BMC biology 2019), the authors already showed translocation formation from the same exact loci using DNA from the same organs of the same transgenic mice with spCas9.

In the revised manuscript, we are using more sophisticated techniques (ddPCR+Duplex-Seq) to more confidently not only detect, but also quantify translocation events for both SpOT-ON Cas9 and SpCas9 (added Figure 3H-I). In addition, as the reviewer can see, the primers used by Carreas et al 2019 were not very specific (non-expected PCR-products obtained; Fig.5F). This motivated re-analyzing the same gDNA, now using more accurate methods also allowing us to quantify the editing.

Minor concerns:

S5: “f” is missing in the legend.

FigS6: no legends on the graph A

Corrected in the revised version of the manuscript.

In conclusion, the SPOTon Cas9 is a promising nuclease but a deeper characterization of the higher specificity is needed to really demonstrate a significant improvement of the targeting. The induction of in frame mutations may be a serious drawback for the use of this Cas9 for KnockOut experiments for example.

The data concerning the decrease in translocation formation do not fully support the conclusions.

We thank the reviewer for recognizing SpOT-ON as a promising nuclease. We have designed and executed a wide range of experiments as described above to address all of the remaining concerns. We believe that the additional studies with in-depth analysis of SpOT-ON have strengthened our understanding of this nuclease.

Reviewer #2 (Remarks to the Author):

Bestas et al. performed a Duplex-seq-based candidate off-target sequencing method to reduce sequencing error and to identify low-frequency off-target mutagenesis induced by CRISPR-Cas9 treatment. Next, the authors performed a database search on TIGFRAM and identified a novel MH0245 Cas9 which belongs to type II-B CRISPR. The MH0245 Cas9 cuts DNA slightly slower than SpCas9 and generates a 3-nt long 5'-overhang. Lastly, the authors delivered the MH0245 Cas9 into a mouse model by an adenoviral vector. They identified greater specificity than SpCas9, and less inter-chromosomal translocation between mouse Pcsk9 on chromosome 4 and the human PCSK9 gene inserted into the Rosa26 locus on chromosome 6. This is a well-written manuscript with properly performed experimental data, and the novel MH0245 Cas9 may have broad applicability once it becomes available to the genome editing community. However, the reviewer has some concerns regarding the manuscript, which are described below.

We thank the reviewer for evaluating our manuscript and for acknowledging that our manuscript is well written with properly performed experimental data, and that our novel SpOT-ON Cas9 could spark great interest in the genome editing community. We have

now adjusted the presentation of our data, elaborated on previous results and performed additional experiments to support our conclusions of SpOT-ON as a Type II-B Cas9 nuclease with no detectable off-targets and reduced translocations.

Major Comments

1) Nomenclature of MH0245 Cas9

Although the authors named the novel type II-B MH0245 Cas9 as "SPOT-ON", this naming would confuse the scientific community, as the authors used the native MH0245 Cas9 amino acid sequence without protein engineering (except for codon optimization and addition of NLS). Like "Cpf1" has been re-named as "Cas12a" to retain the consistency of CRISPR nomenclature, the reviewer strongly recommends maintaining the "Cas9" and adding a prefix of specie origin (or other ID like MH0245) to be consistent with another CRISPR-Cas9s, such as SpCas9 or FnCas9.

P32. "Using blastp with the MH0245 Cas9 protein sequence as query and NCBI nr as database resulted in related proteins sequences from a metagenomic species annotated as Succinatimonas and several Sutterella species. However, the hit with the highest score had only 37% protein sequence identity to MH0245 Cas9. "

When the reviewer performed a blastp search against the non-redundant protein sequences database with a default blastp setting, the Cas9 protein sequence from Parasutterella secunda was identified with 99% identity (Sequence ID: WP_258333620.1). Hence, the MH0245 Cas9 can be named "Parasutterella secunda Cas9" or "PsCas9" for short, unless there is evidence that the MH0245 Cas9 is not derived from Parasutterella secunda.

We thank the reviewer for the insightful comment. The discrepancy between reviewer's investigation and the initially reported similarity score is one of the recent updates in the BLAST databases that was not available during the manuscript preparation. Nevertheless, as the reviewer pointed out, MH0245 Cas9 is not identical to WP_258333620.1 and differs in two residues, therefore it cannot be used as definitive proof of its origin. Given that the scientific community has been open and rather flexible with the nomenclature of other Cas proteins recently, such as CasX (renamed from Cas12e), we would opt for keeping SpOT-ON Cas9 as a name of the protein described in the study.

2) Data presentation on in vitro DNA cleavage activity

Fig. 2F, 2G: In addition to the cleavage kinetics (k_{fast}), please also show the "fraction digested" in the graphs. As shown in Figure S4D or S4G, although the cleavage speed of MH0245 Cas9 was 20-300 seconds slower than SpCas9, the final cutting efficiency reached a similar level at the end. When considering the off-target cleavage risk, absolute digestion efficiency should be indicated.

We appreciate this comment. While we believe that cleavage kinetics is one of the major factors determining editing activity (PMID: 32681021, 36306733), we fully agree that the total digested fraction is also an important factor and agree this data would be valuable to include. Relevant figures and associated text have been updated accordingly (Figures 2F,G, Supplementary figure 4H).

3) Duplex-seq method

P3. "Furthermore, most methods used for the evaluation of editing outcomes, including "Verification of in vivo off-targets" (VIVO) strategy, are limited in their sensitivity by the current detection limit of targeted deep sequencing of 0.1%."

The "VIVO" method is a combination of in vitro CIRCLE-seq to identify candidate sites and in vivo deep sequencing analysis. The Duplex-seq may be an advancement of the targeted deep sequencing analysis part, but it still relies on the identification of candidate sites. Although Duplex-seq is a powerful method for reducing the noise caused by sequencing errors, the searchable area is limited to the pre-determined candidate sites. These limitations should be mentioned in the Discussion.

Figure S2 and S3

If the sensitivity of Duplex-seq is 0.01%, please indicate the data with two digits after the decimal point (i.e., "0.03%" rather than "0.0%").

We agree with the reviewer that the dependence of our off-target analysis on a pre-determined set of candidate sites represents a limitation of the experimental set-up. We have added this consideration to the discussion of our manuscript:

"It should also be noted that the applied off-target analysis strategy despite the improved sequencing sensitivity retains the limitation of relying on a previously determined set of candidate sites meaning that off-target mutations not included in this set will be missed."

We have additionally adjusted the presentation of values in Figure S3 to reflect the sensitivity of the method.

4) Cleavage efficiency comparison between MH0245 Cas9 and SpCas9.

P2. "Once loaded with optimal protospacer length, MH0245 cleaves DNA substrates at a similar rate to SpCas9 (Figure 2G, Supplementary Figure 4J)."

Provided Figure 2G and Supplementary Figure 4J do not have side-by-side comparison data of MH0245 Cas9 and SpCas9. Considering MH0245 Cas9 showed lower genome editing activity in some gRNA targets in HEK293T cells (Fig. 3E), whether the same gRNA sets perform similarly in vitro should be investigated.

We thank the reviewer for this comment. Experiments in Figure 2G (now Figure 2F-G) and Supplementary Figure 4J (now Supplementary Figure 4H-I) were run in parallel using the same reagents, thus, can be compared side by side despite being presented in different panels. However, we agree that this might be confusing for the reader. We acknowledge the lack of SpOT-ON in vitro activity on different DNA substrates. Thus, we performed additional kinetics studies of SpOT-ON with 20 and 22-nt spacer and compared it to SpCas9 at AAVS1 and CD34 target sites. We observed that at both DNA substrates, SpOT-ON performs similarly to SpCas9 when loaded with 22-nt spacer and being compared side by side (Supplementary Figure 4E-F).

We thank the reviewer for the insightful observation about Figure 3E (now Figure 3D) showing that at some sites, SpOT-ON on-target activity is lower than for SpCas9. As described above in the response to reviewer #1, this experiment was performed with 20-nt spacer for all enzymes in order to target the same genomic sites and to not bias

off-target activity outcomes due to additional nucleotides in the spacer. As we showed in Figure 2, the 20-nt spacer is suboptimal for SpOT-ON. Thus, we decided to repeat and expand this experiment and analyze editing properties of SpCas9, FnCas9 and SpOT-ON with both 20 and 22 nt spacers. Figures 3 D-E, text, legends and methods were modified accordingly.

5) Translocation efficiency between Fig. 1 and Fig. 4

Although the on-target editing efficiency was greater in Figure 1 than in Figure 4, the translocation efficiency of SpCas9 in Figure 1 (around 0.16%) was lower than that in Figure 4 (about 0.4-0.6%). Why did this discrepancy happen? If this is due to the artifact of deep sequencing, it is recommended to measure the translocation efficiency of Figure 4 samples by ddPCR, similar to Figure 1.

We thank the reviewer for this question and have now clarified in the manuscript that the ddPCR assay in Figure 1 only shows one possible translocation type while the Duplex-seq data takes all possible combinations into account.

Text removed from the manuscript: "...which is consistent with our previous ddPCR result"

Clarification added to the manuscript: "*The Duplex-sequencing method should cover all possible translocation events, while the ddPCR-assay investigates a subset of them. This is reflected in the slightly higher translocation efficiencies in Figure 4 vs Figure 1.*"

Minor comments

"CRISPR/Cas9" should be "CRISPR-Cas9".

"effective binding of the Cas-sgRNA complex to DNA requires a protospacer adjacent motif (PAM) upstream of the targeted DNA sequence."

The term "upstream" here may not be appropriate for SpCas9 or FnCas9. Please revise (downstream or 3' side).

Figure S6A

Please indicate the graph legend for the black box, red circle, and green triangle.

The inconsistencies and mistakes have been corrected in the revised version of the manuscript.

Reviewer #3 (Remarks to the Author):

Bestas et al. discuss their findings relating to a novel CRISPR-Cas9 variant with lower off-target editing and demonstrate it's effective for in vitro and in vivo editing. Furthermore, they present application of an existing sequencing methodology with improved sensitivity, which is important for further exploration of CRISPR technologies in a clinical setting. The manuscript is well written and provides strong argumentation. However, there are a few comments/suggestions that require more explanation.

We thank the reviewer for the evaluation of our manuscript and for highlighting that our manuscript is well written with a strong argumentation, will resonate well with within the CRISPR community and for their suggestions for clarification to further strengthen our manuscript.

- Page 4: How are significance levels determined between targeted amplicon sequencing and Duplex-seq? Also, how is significance calculated between treated and untreated for Duplex-seq?

We thank the reviewer for raising this important consideration. We think the presence of off-targets in an edited sample cannot be evaluated by statistical test significance, as even if an off-target is detected in one sample only, it should be deemed as a concern. Therefore, we reported all detected off-targets above the assay limit of detection which is 0.1% for Amplicon-seq and 0.01% for Duplex-seq.

- Page 5: What is the sequence homology between MH0245 and SpCas9?

The sequence homology between MH0245 and SpCas9 is 21,32%.

- Page 7: The authors describe mismatch tolerance for positions 6-13 and 15-18. Are there potential mismatches in the queried off-target sites?

Yes, the 5 off-target sites edited in vivo by SpCas9 and not SpOT-ON have mismatches in this position with respect to the on-target sequence.

- Page 7: The authors mention a prevalent indel profile of 3nt, which can be explained by the nature of the overhangs. This finding should be verified over a larger number of target sites/guideRNAs.

We agree with the reviewer that a larger number of target sites will increase the confidence in our observation. We have now tested five additional target sites (*PLN*, *CFTR*, *SERPINA1_1*, *SERPINA_2* (two independent sgRNAs) and *APOE*), all showing 3-nt insertions for SpOT-ON compared to prevalent 1-nt indels for SpCas9. We have now included the additional indel profiles in Supplementary Figure 5B.

- Page 8: Can you please describe the dsDNA oligo in more detail? Are there any homologous arms present as the be able to differentiate between NHEJ and HDR? If not, what is the ability of MH0245 to do HDR?

We thank the reviewer for the question. To clarify the design of the dsDNA donor, we have added a reference to the text describing the blunt-ended dsDNA (Tsai et al. PMID: 25513782). Specifically, the blunt-ended dsDNA donor comprises two annealed short oligos of 34 bp without any homology arms, as adapted from Tsai et al. PMID: 25513782. The staggered dsDNAs contain 3-nt long 5' overhangs complementary to the target site. To increase stability of oligos, we added two phosphorothioate linkages at the 5' and 3' ends of both strands. We have demonstrated that integration of blunt-ended and staggered dsDNAs depends on NHEJ, as inhibiting DNA-PK, a key factor of NHEJ repair, significantly decreased integrations.

We acknowledge the reviewer's comment that we did not address the ability of MH0245 to perform HDR. To further demonstrate SpOT-ONs' ability to direct HDR-mediated repair, we performed additional experiments to introduce disease associated mutations at four target sites. We modelled the following variants using both SpOT-ON and SpCas9:

- F508del mutation in *CFTR* (delCTT), cystic fibrosis
- R14del mutation in *PLN* (delAGA), cardiomyopathy
- E342K mutation in *SERPINA1* (G>A), alpha-1 antitrypsin deficiency (two independent sgRNAs)

We transfected HEK293T with single-stranded oligodeoxynucleotide templates containing the mutated sequence of interest flanked by 50nt homology arms to the respective locus along with SpOT-ON or SpCas9 and corresponding sgRNAs. SpOT-ON introduced the desired mutations in a target-specific manner ranging from around 0.5% to 40%, with a similar efficiency as that observed in SpCas9 transfected cells. We added the corresponding data as Supplementary Figure 5F and have introduced a corresponding paragraph into the manuscript.

- Page 9: The authors should do off-target nomination studies for their novel Cas9-variant. The lower levels at off-target editing using sites defined by SpCas9 look very promising, but it does not rule out that MH0245 can have a distinct off target profile from SpCas9.

- Page 10: If MH0245 has a unique off target profile, putative translocations will not be found with the proposed approach which relies on the SpCas9 off target profile.

We would like to thank the reviewer for the suggestion. In order to compare off-target profiles of both nucleases with an unbiased and sensitive method, we performed CHANGE-seq experiments with SpCas9 and SpOT-ON programmed by the promiscuous HEK4 sgRNA. We identified many off-target cleavage sites for SpCas9 in vitro: 1087, 655 and 651 sites in three technical replicates respectively, of which 372 sites were shared across all replicates (Figure 3G). For SpOT-ON, the number of detected off-target cleavage sites were much lower: 49, 31 and 31 sites in three technical replicates respectively, with only 6 sites being shared across all replicates (Figure 3G, Supplementary Data Table 7). Among these 6 off-target sites, all of them contained NGG PAM and were present in SpCas9 samples.

Reviewers' Comments:

Reviewer #1:

Remarks to the Author:

The comments to the authors are written after "*****"

Reviewer 1

Major concerns:

1-For SpotON Cas9 improved specificity:

The Spot-ON Cas9 has a residual 20% editing efficiency on the hPCSK9 locus. This locus has a single mismatch at position 11 compared to the targeted mPCSK9 locus. Even if the editing efficiency is decreased by ~40% compared to spCas9, this mismatch is located between the position 1-16, position where the SpotON activity is supposed to be drastically reduced as shown by the authors (Fig3D). Thus, we could have expected a greater decrease of editing for this off-target locus.

We thank the reviewer for the valuable comment. Indeed, in our in vivo model SpOT-ON still shows a significant albeit reduced activity at the hPCSK9 locus containing a single mismatch to the gMH guide RNA, whereas in cell culture a single mismatch at a similar position resulted in almost complete ablation of the editing activity as shown in Fig3D. Importantly, these two experiments were performed with different sgRNA spacers, targeting different sites, the PCSK9 and EMX1a, respectively. It was postulated previously (PMID: 33608277) that not only mismatch position, but also sequence context is determining the off-target activity of Cas enzymes. This was further proven structurally in an elegant study from the Jinek lab (PMID: 36306733). Moreover, the gMH guide and the hPCSK9 DNA target form a rU:dG mismatch, which is better tolerated due to the wobble base pairing mechanism. This mechanism allows for editing activity of both enzymes, but with reduced effect for SpOT-ON Cas9 due to its observed higher fidelity. Collectively, the residual activity of SpOT-ON Cas9 at the hPCSK9 locus in the mouse model might seem surprising, but this finding doesn't contradict the presented data and goes along well with existing views on mismatch tolerance in the field. To improve the manuscript clarity for the reader we have :

1. Clearly stated that EMX1a locus was used in Figure 3D

2. Clarified this original text:

"Importantly, while SpCas9 showed similar on-target efficiency at both human and mice loci despite the single G-U mismatch at position 11, SpOT-ON distinguished both loci and edited the human PCSK9 locus ~40% less than the mouse Pcsk9 locus (Figure 4B)."

Updated text:

"Importantly, while SpCas9 showed similar on-target efficiency at both human and mice loci, SpOT-ON distinguished both loci and edited the human PCSK9 locus at ~40% lower efficiency than the mouse Pcsk9 locus despite the rU-dG mismatch that is often tolerated due to wobble base pairing (Figure 4B)."

3. Clarified this original text:

"This increased sensitivity to mismatches in vivo is in line with our in vitro and HEK293T cells observations."

Updated text:

"This improved fidelity in vivo is in line with our observations of increased mismatch tolerance in vitro and in HEK293T cells for EMX1a targeting guide RNA. "

**** I agree that the off target efficiency depends on the target and the design of the gRNA. However it is not clear if the authors are referring to Fig3C and not Fig3D for EMX1 in vitro results?

Thus an overall concern is that the authors should give a deeper characterization of the off-target

effects. The authors focus their analysis on predicted off target sites expected for the spCas9 for only one locus. As SpOT-ON Cas9 is a novel variant, the higher specificity should be assessed through a more genome wide method to make sure that other unpredictable off-target sites are editing and for more than one targeted locus.

We agree with the reviewer. To more extensively characterise the off-target properties of this novel enzyme, we performed CHANGE-seq which is a genome-wide off-target discovery approach (PMID: 32541958). We profiled SpCas9 and SpOT-ON-activities using the promiscuous HEK4 sgRNA. Briefly, we identified many off-target cleavage sites for SpCas9 (1087, 655 and 651 sites in the three technical replicates respectively. Among these, 372 sites were shared across all replicates) (Figure 3G).

In contrast, for SpOT-ON, the number of detected off-target cleavage sites were much fewer: 49, 31 and 31 sites in the three technical replicates respectively, with only 6 sites being shared across all replicates (Figure 3G, Supplementary Data Table 7). Those 6 off-target sites were also present in the SpCas9 samples and all of them contained the NGG PAM.

****The CHANGE-seq results are very valuable for this study.

In terms of efficiency to cleave targeted sites, SPOT-ON Cas9 appears as less efficient (50% less efficient) for 2 out of 4 loci in Fig3E.

We thank the reviewer for the comment. For the experiment presented in Figures 3E and 3F initially, we used the same guide RNA spacer of 20-nt (which is suboptimal for SpOT-ON as we showed in Figure 2). We have chosen this experimental setup to compare the off-target properties of the enzymes guided to the same sequences. However, we realize that using 20-nt spacer guide RNA with SpOT-ON Cas9 might reduce off-target editing due to overall lower activity as shown in Figure 2. Thus, we repeated and expanded this experiment and compared the on- and off-target editing of SpOT-ON, SpCas9 and FnCas9 with both 20 and 22-nt spacers. This updated data set will completely substitute panels D and E in Figure 3. We also modified the text in the manuscript accordingly. In the updated data set we have indeed noticed that at some sites the SpOT-ON Cas9 has lower performance than SpCas9 despite being used with optimal spacer. However, at other targets shown in FigS5A the activity of both enzymes is comparable. We concluded and mentioned it in the text that this discrepancy is likely to be sequence and locus specific, which is a known property of Cas9 enzymes.

**** Even with the optimized length of the spacer (22nt) SpOT-ON Cas9 still shows significant decrease (< 30%) of on target activity for half of the loci compared to spCas9 (FANCF and EMX1), this should be clearly written in the text.

Another major concern is that the SPOT-ON Cas9 induces mostly in frame mutations. This can lead to residual protein expression (as shown in FigS6A) with potential toxic activities.

SpOT-ON generates DNA overhangs as a result of its catalytic activity. Those overhangs will often, but not always, result in three nucleotides insertions among other outcomes of DNA repair. We have noticed that the effect is site-specific and depends on the DNA sequence context. For example, we very often see MMEJ-type of repair at many loci as shown in the updated and expanded Supplementary Figure 5B. In addition 3 nucleotides insertions could result in gene knock-out when targeting introns-exons junctions. We agree that when using SpOT-ON to inactivate genes, the user needs to screen for sgRNA sequences not favouring in frame mutations.

****This should be clearly mentioned in the discussion part.

2- Concerning the Duplex-Seq assay used to detect off-target editing:

One major limiting factor is the design of the probe for all potential off-target sites. Indeed only 75 out of 79 "met technical requirements during probe design". Thus 4 potential off target sites are not evaluated for editing compared to amplicon sequencing.

We thank the reviewer for this question and would like to clarify that the 79 off-target sites that were analyzed in the original amplicon-sequencing experiment did not reflect all potential off-target sites of the gMH sgRNA. In the original VIVO publication, CIRCLE-seq identified 529 potential off-target sites, of which only a subset was chosen for validation by amplicon-sequencing. The aspiration of the present study was not to provide an extensive interrogation of all possible off-target sites, but rather to compare the performance of the amplicon sequencing and Duplex-sequencing pipelines on a subset of off-target sites. Therefore, we are not concerned about the drop-out of the four mentioned sites.

In addition, the amplicon sequencing for OT-2 and OT-4 (Figure 1C) show respectively 0.04% and 0.08% of editing so these off-target sites were already detected (why these numbers are not in red is unclear). For OT-2, the % of edition is even higher by amplicon seq? (0.08% compared to 0.01% by Duplex-Seq)? In addition, the sensitivity is still low since for 3 of the off-target edited loci (OT-1, OT-3 and OT-5) only 1 of the triplicate show editing >0%. In consequence, it appears that the Duplex-seq show only moderate improvement in sensitivity to detect off-target sites.

We thank the reviewer for pointing this out. The data shown in Figure 1C is derived by amplicon sequencing which has a lower limit of detection of 0.1%. Editing events that are measured at frequencies below this threshold (i.e. all events measured at frequencies <0.1%) are non-distinguishable from random sequencing error-induced events, and therefore cannot confidently be classified as real editing events. The reported values of 0.04% and 0.08% thus fall below this threshold. Therefore, they have not been colored red and cannot directly be compared to the Duplex-seq-derived data. In contrast to amplicon sequencing, the frequencies obtained by Duplex-seq are above the respective sensitivity threshold of the method (i.e. 0.01%) allowing us to regard these events as actual editing events. The advantage of the 10-fold increase in sensitivity achieved by Duplex-seq is thus the ability to more confidently distinguish real off-target events from sequencing-induced background. The reason why not all detected off-target sites were identified to the same degree in all three animals is likely to be natural variability that is always present in in vivo studies. While conditions in in vitro experiments are more controllable and reproducible, small physiological differences between individual mice might have an impact on the editing outcome leading to detectable editing in some animals while others might only show below-threshold editing.

****Thank you for clarifying.

3- Translocation formation after SPOTon Cas9 cleavage:

Overall, more data are needed to formally prove that chromosomal translocation is decreased upon SPOTONCas9 cleavage. In addition, the method of detection by ddPCR of translocations is probably not the most suitable to detect these events in mouse cells:

In a mechanistic point of view, there is not rational to support the idea that SPOTon Cas9 will induce less translocations (see below).

The authors should look at chromosomal translocation formation in their in vitro model between several loci (they have 4 edited loci) to demonstrate that SPOTon Cas9 cleavage is less prone to form translocation.

Concerning the in vivo data, the authors did not strictly demonstrate that the translocation formation is reduced with the SPOTon Cas9. Translocation frequency is linked to the cutting efficiency. Since the hPSC9 editing efficiency is decreased by 40% when using SPOTon Cas9, the translocation frequency will be decreased as well, and not only because the SPOTon Cas9 lead to less translocations.

We thank the reviewer for the suggestion to further investigate translocation rates. To address this concern, we have performed additional experiments in HEK293T cells and measured the chromosomal translocations by ddPCR. We co-transfected cells with two gRNAs that were previously shown to induce translocations, and used established ddPCR assays (PMID: 33479216, (<https://doi.org/10.1101/2022.12.15.520396>) to measure balanced translocation events between HIST1H2BC-HBEGF and PCSK9-HBEGF loci. We measured the on-target single gRNA editing

efficiencies of both SpCas9 and SpOT-ON at all loci with amplicon-seq. Since both variants were similarly efficient at the HIST1H2BC and PCSK9 loci, we used the relative editing efficiency at the HBEGF locus to normalize the raw translocation frequencies measured by ddPCR to account for the discrepancy in cutting efficiency at certain loci mentioned by the reviewer. Even after normalization, we found that SpOT-ON drastically is associated with reduced translocation frequencies between simultaneously cleaved sites and have added these experiments in Figure 3 H-I with a corresponding text in the Results section.

**** It seems that HBEGF editing efficiency is the same for both Cas9 (SuplFig5?) so it is not clear why the authors use this loci to normalize. In the contrary I don't find the quantification of the editing efficiency for HIST1H2BC loci.

In addition, the quantification of translocations by droplet PCR needs accurate settings and is not very sensitive. It is possible to compare translocation frequency with ddPCR but it does not directly answer the question of the formation of translocation after SPOTON Cas9 cleavage. In the contrary for detection of low translocation frequency, a simple PCR done on DNA serial dilutions is more accurate to detect translocated events at a frequency $< 10^{-6}$.

We would like to clarify that here we aimed at quantifying rather than detecting translocations. Regular PCR would not allow us to compare the frequency at which translocations happen after genome editing executed by SpOT-ON or SpCas9. In contrast to regular PCR, Droplet Digital PCR (ddPCR) is a quantitative and more precise method. Instead of providing an end-point result from a single amplification event for each sample, ddPCR allows $\sim 15\,000$ - $20\,000$ individual amplification reactions to take place per sample, and most of these contain only one template, avoiding any bias from preferential amplification of shorter products seen in regular PCR. Also, it is more precise than regular PCR due to a lower risk of off-target amplification as not just the primers, but also the probe, needs to bind DNA (when using probe-based ddPCR-assays as we did here).

In fact, several publications have described ddPCR as a highly sensitive method, for those by Kojabad et al. PMID 33538349, Campomenosi et al, Hindson et al. PMID 23995387, and RT-ddPCR was identified as more sensitive than qPCR e.g. in Robinson et al, PMID 29506703, and Tsujimoto et al, PMID 36378656.

****ok

Furthermore, the ddPCR requires very small amplicons (< 100 bp) which are not suitable for accurate translocation formation detection. Deletions/insertions are induced at breakpoint particularly in mouse cells (which are mostly using altNHEJ to repair translocation breakpoints) (Reviews: Ramsden et al, 2021, DOI: 10.1038/s41388-021-01856-9; Brunet and Jasin, 2018, DOI: 10.1007/978-981-13-0593-1_2). The lack of detection of "modified" breakpoints may artificially decrease the translocation frequency obtained by ddPCR. As described by the authors, SPOTON induced insertions.

We thank the reviewer for pointing this out and want to clarify the design of our ddPCR experiments. We carefully designed the ddPCR-assay to ensure we would detect translocations even if indels form around the break points.

From the methods section: "In an effort to obtain an as accurate result as possible, the primers and probes used in the ddPCR assay were designed to bind in an area at least 50 bp upstream and downstream of the cut site/nickase site (thereby avoiding placing them close to the cut site where potential indels could have prevented the primers or probe from binding, resulting in an underestimated translocation frequency)."

****Ok

In their previous study (Carreras et al, BMC biology 2019), the authors showed high level of formation of the acentric translocated chromosome that is not quantified when using the SPOT ON Cas9.

The experimental procedure of the Duplex-Seq set-up in fact allows detection of all four possible combinations of translocations between the mouse and human PCSK9 loci (i.e. acentric, dicentric, as well as the two balanced translocations). We have added the data for both SpCas9 and SpOT-ON (Supplementary Figure 6C-D). While the absolute number of overall translocations was significantly lower for SpOT-ON vs. SpCas9, both enzymes showed a similar distribution of detected translocations with dicentric and acentric translocations being most common.

****There is no Supl Fig 6C and D? I guess it is referring to Supl Fig7C and D?

In the second paragraph the authors wrote the sentence " While the mechanism of CRISPR/Cas9-induced DNA translocations remains largely elusive, the blunt-ended DNA formed after the SpCas9 cut seems to promote this process », this sentence is inaccurate since numerous studies have deciphered in details the molecular mechanism of chromosomal translocations both in human and mouse cells (for reviews: Ramsden et al, 2021, DOI: 10.1038/s41388-021-01856-9; Brunet and Jasin, 2018, DOI: 10.1007/978-981-13-0593-1_2). Similarly, the sentence "its nuclease activity generated 3-nt long 5'-overhangs in the DNA substrate, which suggests that this Cas9 nuclease may decrease the risk of translocations » is misleading: translocations can be easily induced by the use of TALENs and Zinc Finger Nucleases that typically lead to 5'overhangs formation. The authors do not provide any references for their statements...

We thank the reviewer for insightful comments and for pointing out critical references that we unfortunately missed.

These misleading sentences have now been removed from the manuscript:

- "While the mechanism of CRISPR/Cas9-induced DNA translocations remains largely elusive, the blunt-ended DNA formed after the SpCas9 cut seems to promote this process. In fact,..."
- "which suggests that this Cas9 nuclease may decrease the risk of translocations"

ok

Other concern: it is unclear what new results are described in the Figure 1 E and F (main Figure) compared to the previous paper of the same authors. In this former paper (Carreras et al, BMC biology 2019), the authors already showed translocation formation from the same exact loci using DNA from the same organs of the same transgenic mice with spCas9.

In the revised manuscript, we are using more sophisticated techniques (ddPCR+Duplex-Seq) to more confidently not only detect, but also quantify translocation events for both SpOT-ON Cas9 and SpCas9 (added Figure 3H-I). In addition, as the reviewer can see, the primers used by Carreras et al 2019 were not very specific (non-expected PCR-products obtained; Fig.5F). This motivated re-analyzing the same gDNA, now using more accurate methods also allowing us to quantify the editing.

****ok

**** In conclusion, the additional experiments really improved the quality of the study (particularly with the CHANGE-seq results.

Please make sure to verify all the numbering/references of the Figures.

Please answer this point:"It seems that HBEGF editing efficiency is the same for both Cas9 (SuplFig5?) so it is not clear why the authors use this loci to normalize. In the contrary I don't find the quantification of the editing efficiency for HIST1H2BC loci".

and add sentences

- concerning the decrease of editing efficiency for some targeted loci with SPOTON (Even with the optimized length of the spacer (22nt) SpOT-ON Cas9 still shows significant decrease (< 30%) of on target activity for half of the loci (FANCF and EMX1), this should be clearly written in the text)

-concerning in frame mutations in the discussion.'We agree that when using SpOT-ON to inactivate genes, the user needs to screen for sgRNA sequences not favouring in frame mutations'

Reviewer #2:

Remarks to the Author:

The manuscript has been revised, but there are still a few items that need to be corrected. Especially, nomenclature of MH0245 Cas9 is an important issue to avoid confusion for the readers. Specific comments are as follows.

Response letter: "The discrepancy between reviewer's investigation and the initially reported similarity score is one of the recent updates in the BLAST databases that was not available during the manuscript preparation."

> Please update the description in Lines 775-781, based on the latest BLAST database.

Response letter: "Nevertheless, as the reviewer pointed out, MH0245 Cas9 is not identical to WP_258333620.1 and differs in two residues, therefore it cannot be used as definitive proof of its origin."

> If 99% amino acid identity is not regarded as evidence of its origin, what else is needed? The manuscript should report the latest findings based on the updated database information, and it is fully the authors' responsibility to provide as comprehensive information as possible regarding the origin of newly identified Cas9.

Line 150: "MH0245 Cas9 shares a high homology to WP_258333620.1 from *Parasutterella secunda*."

> Please include the percentage of amino acid identity.

Response letter: "Given that the scientific community has been open and rather flexible with the nomenclature of other Cas proteins recently, such as CasX (renamed from Cas12e), we would opt for keeping SpOT-ON Cas9 as a name of the protein described in the study."

>The reviewer thinks "SpOT-ON (Specific Off-Target/ON-target) Cas9" is not an acceptable name for a gene as it is confusing with other existing Cas9 homologs. "Specificity" is a relative measure and the authors only compared the specificity against SpCas9 in this manuscript, so "more specific Cas9" can exist from other origins.

Moreover, "SpOT-ON" must not be used as an alternative for "Cas9", since "Cas (CRISPR associated)" terminology is commonly accepted. The authors should consider this issue seriously.

Response letter: "While we believe that cleavage kinetics is one of the major factors determining editing activity (PMID: 32681021, 36306733), we fully agree that the total digested fraction is also an important factor and agree this data would be valuable to include. Relevant figures and associated text have been updated accordingly (Figures 2F,G, Supplementary figure 4H)."

> According to the new Sup Figure 4H, MH0245 Cas9 cleaves nearly 100% of target DNA in vitro, despite single base mismatch (except for the position 6, 16-18). This is important information for the readers, so should be included in the main Figure 2.

Also, in Sup Figure 4J, SpCas9 cleaves target DNA even with one base pair mismatch at any position, including the PAM sequence. This data contradicts from previous reports, but can the authors comment on why that is?

"We have additionally adjusted the presentation of values in Figure S3 to reflect the sensitivity of the method."

> Similar to the above, please consider the sensitivity of Figure S2. If the detection limit is 0.1%, on-target cleavage activities should be presented as one digit after the decimal point (i.e., "46.1%" rather than "46.08%").

Reviewer #3:

Remarks to the Author:

The authors addressed the suggestions of the first draft, and I have no further comments on their current manuscript.

Response to reviewers

Bestas B. et al. - SPOT-ON is a Cas9 nuclease with no detectable off-targets and reduced chromosomal translocations in vivo.

Thank you for reviewing our manuscript for publication in Nature Communications, and for your additional suggestions regarding how to improve the revised version. As a general note, in the revised version, we refer to MH0245 Cas9 / SpOT-ON Cas9 as to PsCas9 as per Reviewer's 2 suggestion. Below we outline our response to the comments.

REVIEWER COMMENTS

Reviewer #1 (Remarks to the Author):

1. I agree that the off target efficiency depends on the target and the design of the gRNA. However it is not clear if the authors are referring to Fig3C and not Fig3D for EMX1 in vitro results?

We apologise for the mistake and the confusion. The figure reference has been corrected.

2.The CHANGE-seq results are very valuable for this study.

We appreciate your previous suggestion and want to thank you for your kind comment.

3. Even with the optimized length of the spacer (22nt) SpOT-ON Cas9 still shows significant decrease (< 30%) of on target activity for half of the loci compared to spCas9 (FANCF and EMX1), this should be clearly written in the text.

We agree with the reviewer's comment and acknowledge the fact that the editing efficiency is lower on some loci than for SpCas9. Both suggestion #3 and #4 have been added to the manuscript, please see below (#9) for details.

4.This should be clearly mentioned in the discussion part.

We agree with the reviewer's comment regarding the DNA overhangs and ability to introduce in frame mutations. Both suggestion #3 and #4 have been added to the manuscript, please see below (#9) for details.

5.There is no Supl Fig 6C and D? I guess it is referring to Supl Fig7C and D?

Apologies for the mistake. We have now carefully reviewed all figure references and corrected them where needed.

6. In conclusion, the additional experiments really improved the quality of the study (particularly with the CHANGE-seq results).

We thank the reviewer for suggesting to include CHANGE-seq, and for recognizing our efforts to improve the manuscript's quality.

7.Please make sure to verify all the numbering/references of the Figures.

All the figures in the revised manuscript have now been now cross-checked.

8. Please answer this point: "It seems that HBEGF editing efficiency is the same for both Cas9 (SuplFig5?) so it is not clear why the authors use this loci to normalize. In the contrary I don't find the quantification of the editing efficiency for HIST1H2BC loci".

Thank you for pointing this out. We realize that performing this normalization is not necessary and are now showing the translocation results without normalizing to the editing efficiencies at the HBEGF locus. We removed the text related to this from the Result section, Fig 3-legend and Method section in the manuscript. The differences in translocation efficiencies between SpCas9 and PsCas9 are very pronounced regardless of whether normalization was performed or not (please see the graphs below).

We have updated Fig 3H-I to show the data without normalization.

With normalization (old version, removed from the manuscript):

Without normalization (new version):

From the Results section submitted in April 2023:

SpOT-ON induces fewer translocations after gene editing in mammalian cells

Chromosomal translocations may occur after induction of multiple, simultaneous DSBs, e.g. between on- and off-target sites cleaved by promiscuous enzymes. Such genomic instability poses a safety liability for using nucleases in clinical applications. Therefore, we investigated if the 5'-DNA overhangs after cleavage by PsCas9 are as translocation-prone as the blunt ends typically generated by wild-type SpCas9. To this end, we transfected HEK293T cells with either SpCas9 or PsCas9 and a pair of sgRNAs previously shown to induce balanced translocations between the two on-target sites by SpCas9^{68,69}. Amplicon sequencing was used to measure the on-target editing efficiency of SpCas9 and SpOT-ON at each target site in single gRNA conditions. Both variants had similar on-target efficiencies at the

~~*HIST1H2BC* and *PCSK9* sites, and we therefore used the efficiency at the *HBEGF* site to normalize the translocation rates.~~ Using ddPCR, we found that PsCas9 induced 22-fold and 11-fold fewer translocations than SpCas9 between *HIST1H2BC* and *HBEGF*, and *PCSK9* and *HBEGF*, respectively (Figure 3H-I). Our results demonstrate that PsCas9 is less prone than SpCas9 to induce translocations after introducing two independent DSBs in the human genome.

9. and add sentences

- concerning the decrease of editing efficiency for some targeted loci with SPOTON (Even with the optimized length of the spacer (22nt) SpOT-ON Cas9 still shows significant decrease (< 30%) of on target activity for half of the loci (FANCF and EMX1), this should be clearly written in the text)

The following sentence has been added in the discussion section:

PsCas9 outperformed wild-type SpCas9 in terms of editing specificity, yet showed similar activity on many sites (HEK3, HEK4, CD34, HBEGF, STAT1 AAVS1) but also compromised activity on several others (FANCF and EMX1).

-concerning in frame mutations in the discussion. 'We agree that when using SpOT-ON to inactivate genes, the user needs to screen for sgRNA sequences not favouring in frame mutations'

The following sentence has been added to the discussion section:

Of note, when applying PsCas9 to inactivate genes, we advise screening for sgRNAs that do not favor in frame mutations.

Reviewer #2 (Remarks to the Author):

The manuscript has been revised, but there are still a few items that need to be corrected. Especially, nomenclature of MH0245 Cas9 is an important issue to avoid confusion for the readers. Specific comments are as follows.

> Please update the description in Lines 775-781, based on the latest BLAST database.

The following sentence has been modified in the materials and methods section:

Using blastp with the protein sequence as query and NCBI as database, resulted in identification of a very high homology to WP_258333620.1 (99%) from *Parasutterella secunda*.

> If 99% amino acid identity is not regarded as evidence of its origin, what else is needed? The manuscript should report the latest findings based on the updated database information, and it is fully the authors' responsibility to provide as comprehensive information as possible regarding the origin of newly identified Cas9.

We acknowledge the reviewer's opinion and accordingly replaced MH0245/SpOT-ON with *Parasutterella secunda* Cas9 (PsCas9) throughout the revised manuscript.

We identified a Cas operon that consists of a genomic architecture characteristic to type II-B, harboring *cas4* and lacking *csn2* (Figure 2A)^{41,42}. The protein product of the *cas9* gene shares a high homology to WP_258333620.1 from *Parasutterella secunda* (99%), therefore we referred to it as PsCas9. Alignment of PsCas9 with all known Cas proteins classified this new member into the type-II-B family (Figure 2B).

> Please include the percentage of amino acid identity.

The following sentence has been modified in the main text:

The protein product of the *cas9* gene shares a high homology to WP_258333620.1 from *Parasutterella secunda* (99%),

>The reviewer thinks "SpOT-ON (Specific Off-Target/ON-target) Cas9" is not an acceptable name for a gene as it is confusing with other existing Cas9 homologs. "Specificity" is a relative measure and the authors only compared the specificity against SpCas9 in this manuscript, so "more specific Cas9" can exist from other origins. Moreover, "SpOT-ON" must not be used as an alternative for "Cas9", since "Cas (CRISPR associated)" terminology is commonly accepted. The authors should consider this issue seriously.

We agree with the reviewer's opinion and will refer to MH0245/SpOT-ON as *Parasutterella secunda* Cas9 (PsCas9) throughout the revised manuscript. Please see the reply above.

> According to the new Sup Figure 4H, MH0245 Cas9 cleaves nearly 100% of target DNA in vitro, despite single base mismatch (except for the position 6, 16-18). This is important information for the readers, so should be included in the main Figure 2.

Indeed, MH0245 Cas9 (PsCas9) can process the substrate almost to completion at the latest timepoint despite the presence of mismatches in the target sequence. However, we do not find these results unexpected. Several previous reports measuring Cas9 kinetics in vitro documented near complete digestion of various imperfect substrates (PMID 26524520, 32681021, 32895548, 36306733). Importantly, substrates with up to 6 mismatches can still be cleaved close to 100% provided enough time, as shown in recent work from Jinek lab (PMID 36306733).

In depth biochemical studies of Cas9 nuclease activity by Johnson lab led to the proposal of a kinetic model that we used in our study (PMID 29320733). In their follow up work, Lie et al. showed that slower DNA cleavage is the main determinant for the higher fidelity of Cas9 enzymes (PMID 32681021). A study from Jinek lab mentioned above (PMID 36306733) also postulates that kinetics is one of the major determinants for the off-target activity. Thus, we decided to use reaction speed for description of PsCas9 and propose it as a factor defining its high-fidelity properties.

Nonetheless, we agree that the cleaved fraction at the latest timepoint might be of interest to the curious reader but might also be confusing. To further highlight the differences between SpCas9 and PsCas9 in DNA cleavage in vitro we will provide an additional panel showing the fraction of DNA digested at 10 seconds alongside with the latest available 10-minute timepoint.

We strongly believe that the most comprehensive yet concise way of presenting our in vitro data will be to place reaction speed panels into the main text while keeping 10' and 10'' fraction cleaved panels in the supplementary information.

Collectively, the following changes have been made:

- Added panels showing mismatched DNA fraction digested for both enzymes at 10 seconds and 10 minutes to Supplementary Figure 4.
- Modified the text of the "**PsCas9 performs best with ≥ 22 -nt spacers and requires specific sgRNA-target pairing**" section to acknowledge the changes above

Also, in Sup Figure 4J, SpCas9 cleaves target DNA even with one base pair mismatch at any position, including the PAM sequence. This data contradicts from previous reports, but can the authors comment on why that is?

We thank the reviewer for the comment. In the Supplementary Figure 4J (and also in Figure 2H and Supplementary Figure 4H, 4I) the numbering corresponds to the nucleotide position upstream of PAM in 3'-to-5' direction. This unconventional counting is common for abbreviating Cas9 mismatch positions in the field, though (ie PMID 32895548, PMID 35236982). Thus, positions 21 to 23 are the furthest from the PAM and affect the kinetics minimally which goes along with existing reports. We explicitly mentioned the abbreviation in the figure legend and referred the reader to the schematic in Supplementary Figure 4G showing the mismatch position and type. To further improve the clarity of these data, we modify the figure legend: **Mismatch positions are labelled in 3' to 5' direction, with Position 1 being directly upstream of PAM.** We also made similar adjustments to the legends of Sup Figures 4H, I, J.

> Similar to the above, please consider the sensitivity of Figure S2. If the detection limit is 0.1%, on-target cleavage activities should be presented as one digit after the decimal point (i.e., "46.1%" rather than "46.08%").

Corrected in the revised manuscript.

Reviewer #3 (Remarks to the Author):

The authors addressed the suggestions of the first draft, and I have no further comments on their current manuscript.

We thank the reviewer for providing insightful feedback through the revision process.